# MS CETSA deep functional proteomics uncovers DNA repair programs leading to gemcitabine resistance

Ying Yu Liang[1], Khalidah Khalid[1], Hai Van Le[1], Hui Min Vivian Teo[2], Mindaugas Raitelaitis[3], Marc-Antoine Gerault[3], Jane Jia Hui Lee[2], Jiawen Lyu[3], Allison Chan[4], Anand Devaprasath Jeyasekharan[4,5,6], Wai Leong Tam ●[2,4,5,7] ✉, Pär Nordlund ●[1,3] ✉ & Nayana Prabhu ●[1] ✉

Mechanisms for resistance to cytotoxic cancer drugs are dependent on dynamic changes in the biochemistry of cellular pathways, information which is hard to obtain at the systems level. Here we use a deep functional proteomics implementation of the Cellular Thermal Shift Assay to reveal a range of induced biochemical responses to gemcitabine in resistant and sensitive diffuse large B cell lymphoma cell lines. Initial responses in both, gemcitabine resistant and sensitive cells, reflect known targeted effects by gemcitabine on ribonucleotide reductase and DNA damage responses. However, later responses diverge dramatically where sensitive cells show induction of characteristic CETSA signals for early apoptosis, while resistant cells reveal biochemical modulations reflecting transition through a distinct DNA-damage signaling state, including opening of cell cycle checkpoints and induction of translesion DNA synthesis programs, allowing bypass of damaged DNA-adducts. The results also show the induction of a protein ensemble, labeled the Auxiliary DNA Damage Repair, likely supporting DNA replication at damaged sites that can be attenuated in resistant cells by an ATR inhibitor, thus re-establishing gemcitabine sensitivity and demonstrating ATR as a key signaling node of this response.

Cancer cells evade cytotoxic drugs through activation of resistance mechanisms. While drug-sensitive cancer cells induce cellular programs leading to cell death, a wide range of cellular processes have been implied in resistant cells, including modulations of drug transport[1] or activation[2], induction of apoptosis blockade[3], bypass of oncogene inhibition by mutations of drug binding site[4], activation of parallel driver pathways[5], as well as modulation of tumor microenvironment[6] and cell-to-cell signaling[7,8]. It is likely that multiple resistance mechanisms are established simultaneously to overcome drug action.

Detailed insights into which resistance promoting programs are operating in cancers of individual patients at different stages of

[1]Institute of Molecular and Cell Biology (IMCB), Agency for Science, Technology and Research (A*STAR), 61 Biopolis Drive, Proteos 138673, Singapore. [2]Genome Institute of Singapore (GIS), Agency for Science, Technology and Research (A*STAR), 60 Biopolis Street, Genome 138672, Singapore. [3]Department of Oncology and Pathology, Karolinska Institutet, 171 77 Stockholm, Sweden. [4]Cancer Science Institute of Singapore, National University of Singapore, 14 Medical Drive, Singapore 117599, Singapore. [5]Department of Biochemistry, Yong Loo Lin School of Medicine, National University of Singapore, 8 Medical Drive, Singapore 117596, Singapore. [6]Department of Haematology-Oncology, National University Cancer Institute, Singapore 119074, Singapore. [7]NUS Center for Cancer Research, Yong Loo Lin School of Medicine, National University Singapore, 14 Medical Drive, Singapore 117599, Singapore. ✉e-mail: tamwl@gis.a-star.edu.sg; par.nordlund@ki.se; nayana_prabhu@imcb.a-star.edu.sg

therapy could arguably be transformative for selection of optimal drug combinations and staging in personalized therapy, as well as for identifying novel drug targets to attenuate resistance responses. However, conclusive elucidation of cancer drug resistance mechanisms is often challenging when they can involve complex remodeling of cellular pathways. Typically, resistance mechanisms are addressed using genomic or transcriptomic approaches, most often assessing static differences between cancer patient samples or sensitive and resistant cancer cells in model systems[9–11]. Although such studies can access key mutations and RNA level changes implicative of resistance, cellular pathways and processes are highly regulated at the biochemical level, information only indirectly accessed using these methods. Moreover, comparison of static cells does not address drug-induced cellular responses which can play key roles in resistance to cytotoxic cancer drugs but are normally not activated during ambient cancer cell growth. Notably, some cancer drug-induced resistance responses can be efficiently studied with focused assays, such as the induction of reactive oxygen species, autophagy and chaperone activation. While useful, these studies require a priori knowledge on putative mode of resistance mechanisms, and do not provide an unbiased view on sequences of regulatory events.

Here, we examined the induced modulation of cellular biochemistry leading to resistance towards one of the more commonly used cytotoxic cancer drugs – gemcitabine, an anti-neoplastic pyrimidine analog that replaces cytidine during DNA replication and inhibits ribonucleotide reductase (RNR)[12]. In various cancers including pancreatic, breast, ovarian, non-small cell lung cancer and lymphoma, gemcitabine is employed either in the first-line or refractory setting. Diffuse large B-cell lymphoma (DLBCL) represents the most frequently occurring and aggressive form of non-Hodgkin's lymphoma. The anthracycline-based regimen R-CHOP (rituximab, cyclophosphamide, doxorubicin, vincristine and prednisone) is the standard of care for first-line treatment with ~60% of the patients achieving complete response[13]. However 20-50% of patients do not respond, or relapse within the first two years of treatment[14]. Gemcitabine has recently been used in salvage regimens for DLBCL although resistance often develops[15]. Gemcitabine is a nucleotide prodrug that needs to be metabolized into its active phosphorylated form within cells to exert its effects (Fig. 1A). RNR catalyzes the conversion of ribonucleoside diphosphates to deoxyribonucleoside diphosphates and is a major protein target for gemcitabine[16,17]. In its diphosphate form, gemcitabine inhibits RNR by forming a covalent adduct to the catalytic subunit (RRM1), or alternatively scavenging the free radical cofactor of RNR, thus depleting dNTP pools[18]. While genomic and transcriptomic studies have helped identify driver pathways and prognostic gene signatures in DLBCL[19,20], this information remains of limited utility in guiding treatment regimens, especially in stratifying patients who may respond to specific salvage therapy agents.

To better understand how sensitive or resistance biochemical pathways become selectively activated in response to therapeutics in DLBCL cells, we apply a time-dependent implementation of the deep functional proteomics method, IMPRINTS-CETSA (Integrated Modulation of Protein Interaction States - Cellular Thermal Shift Assay) to study gemcitabine-induced programs. CETSA reports on modulations of pathway activation at the biochemical level in intact cells by monitoring changes in protein interaction states, i.e., interactions made by individual proteins to other molecules in live cells reflecting protein activity and functional states[21]. MS-CETSA (Mass Spectrometry-based CETSA) is the first integrative technology that can directly assess such protein interaction states in intact cells and tissues but has not been used previously for deep characterization of induced drug resistance. In the present study, we reveal comprehensive and distinct information on the time-dependent biochemical responses of gemcitabine in sensitive and resistant DLBCL cells. Initial responses in both cell types reveal similar RNR inhibition and activation of DNA-damage signaling. However, the downstream response in sensitive cells reflects the characteristic CETSA signature for apoptosis induction[22], while in resistant cells we observe cell cycle progression, translesion DNA synthesis (TLS) and describe the induction of a protein ensemble that likely support DNA repair. This response provides a rationale for gemcitabine resistance in DLBCL cells, which can be reversed by attenuating the DNA-repair inducing pathway with an ATR (ataxia telangiectasia and Rad3-related protein) inhibitor, and thereby re-establishing gemcitabine sensitivity. This study validates IMPRINTS-CETSA[23] as an efficient approach to dissect induced cancer drug resistance pathways at the biochemical level and provide drug targets and biomarkers for combination therapies with potential applications in the clinic.

## Results

To study gemcitabine-induced resistance mechanisms we first evaluated the cell viability of a panel of DLBCL cell lines when challenged with a range of gemcitabine doses over 48 h. Of the profiled cell lines, we selected two sensitive (OCI-LY19, $IC_{50} = 2.4$ nM and OCI-LY3, $IC_{50} = 14.4$ nM) and two resistant (SUDHL4, $IC_{50}$ = not defined and HT, $IC_{50}$ = not defined) cell lines (Supplementary Fig. 1A) to employ the highly sensitive IMPRINTS-CETSA format, whereby 3 biological replicates of treated cells were labeled together with their vehicle controls. To capture the dynamic cellular response upon drug treatment, sensitive OCI-LY19 and resistant SUDHL4 cells were treated for 4 time points (1 h, 3 h, 5 h and 8 h), while for comparison purposes sensitive OCI-LY3 and resistant HT cells were treated only at 2 timepoints (1 h and 8 h). For informative CETSA responses to be measurable, the drug concentration needs to be sufficiently high to induce molecular perturbations with high stoichiometry. We therefore selected 20X $IC_{50}$ for the sensitive cells, i.e. 48 nM for OCI-LY19 and 288 nM for OCI-LY3. For both resistant cells we used 500X the concentration in relation to OCI-LY19, i.e. 24 μM, but have additionally collected CETSA data at lower concentrations (480 nM and 48 nM) to monitor dose-dependent responses. We confirmed that treated cells remained intact and viable, as judged from a trypan blue assay, at the maximum timepoint for the CETSA experiments (Supplementary Fig. 1B). IMPRINTS-CETSA was performed similarly in all cell lines using a 6-temperature protocol (Fig. 1B) with the interpretation of IMPRINTS profiles explained in Fig. 1C. CETSA is based on the biophysical concept that, with increasing temperatures, proteins denature and precipitate out of the soluble fraction resulting in distinct melting profiles for each protein. Changes in the protein interaction states through e.g. binding to a ligand (e.g. drug), interaction with other molecules (e.g. proteins, DNA, RNA, metabolites), or posttranslational modifications, can lead to a shift in the melting profile, which is observed as thermal stabilization or destabilization. An IMPRINTS-CETSA profile illustrates the difference in the abundance of measured soluble protein between vehicle and treatment conditions at a given temperature. The protein coverages and numbers of hits scored using our standard hit selection criteria (described in Materials & Methods) are shown in Supplementary Data 1.

The association of gemcitabine with RNR was expected to increase protein stability and produce a thermal shift. Indeed, within 1 h, the large and catalytic subunit, RRM1, displayed similar IMPRINTS profiles in both resistant and sensitive DLBCL cells, supporting extensive target engagement and inhibition of de novo deoxyribonucleotide synthesis (Fig. 1D). This thermal stabilization was also seen at subsequent timepoints of 3 h, 5 h, and 8 h. An isothermal dose response (ITDR) experiment also showed comparable dose-dependent stabilization of RRM1, supporting similar target engagement in sensitive and resistant cells (Fig. 1E).

### Initiation of DNA damage response in sensitive and resistant cells

Apart from RNR inhibition, gemcitabine acts by being incorporated into DNA, inducing single strand DNA (ssDNA) breaks and stalled

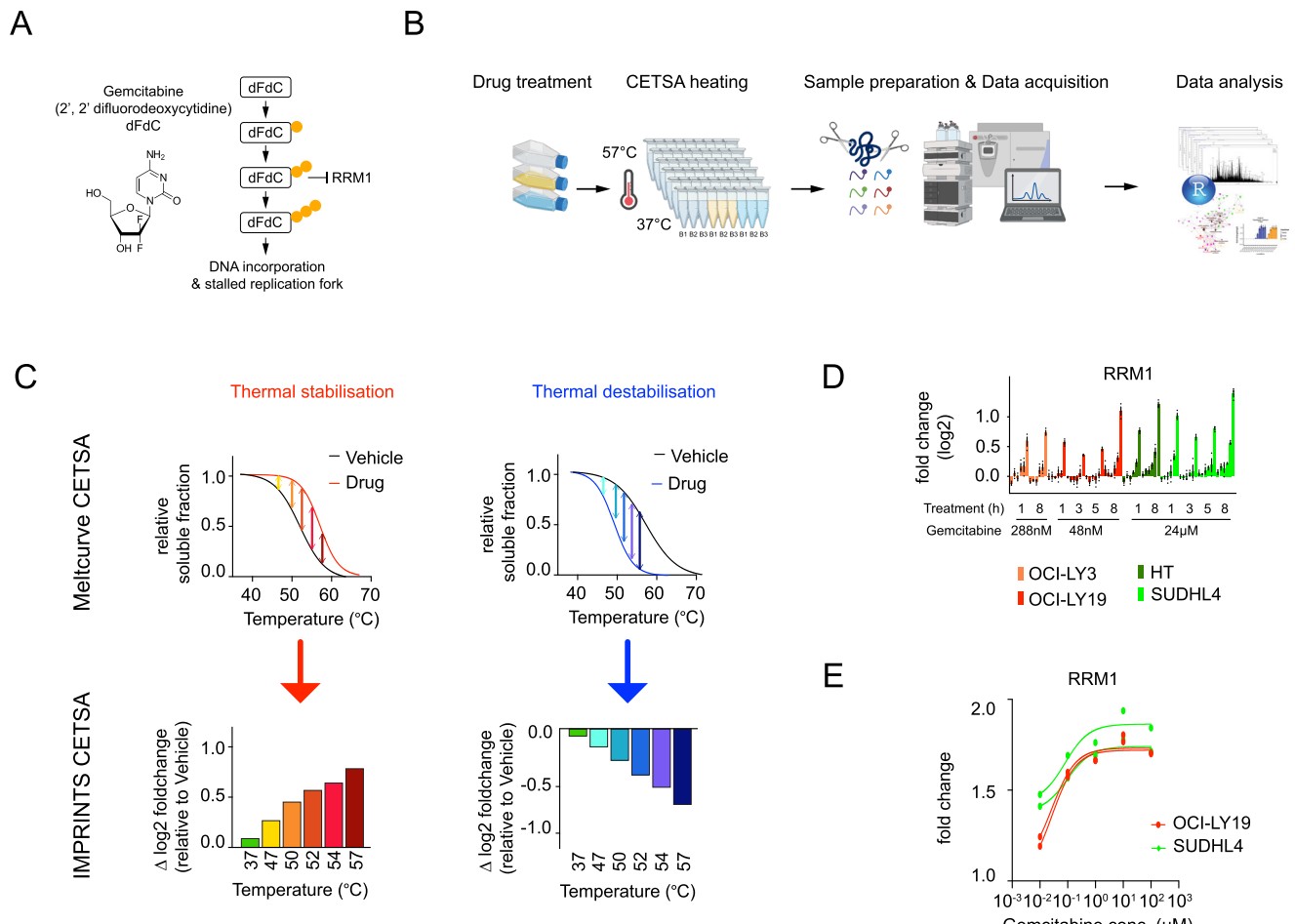

**Fig. 1 | Gemcitabine drug target-engagement in sensitive and resistant cells.** **A** Structure and thus far known MoA of gemcitabine. **B** IMPRINTS CETSA experimental workflow. Created in BioRender. Tam, W. (2025) https://BioRender.com/dufluq1. **C** Interpretation of IMPRINTS CETSA profiles. **D** IMPRINTS CETSA profiles of RRM1 in two sensitive, OCI-LY3 (orange) and OCI-LY19 (red), and two resistant, HT (dark green) and SUDHL4 (light green), cell lines, after 1 h, 3 h, 5 h or 8 h of gemcitabine treatment. Data are presented as mean log2 fold change compared to the reference ±SEM from biological replicates ($n = 3$). Source data are provided as a Source Data file. **E** 3 h Isothermal Dose Response (ITDR) of RRM1 in OCI-LY19 (red) and SUDHL4 (green) cells with different doses of gemcitabine and at 52 °C CETSA heating. Data are presented as mean fold change compared to the reference from technical replicates ($n = 2$). Source data are provided as a Source Data file.

replication forks[24]. Indeed, five proteins - RPA1, RPA2, RPA3, CHEK1 and DNMT1 (Fig. 2) - located at replication sites showed similar CETSA profiles across the various time points and cell lines (Fig. 2). Out of these, 4 belong to the core molecular machinery for sensing and signaling ssDNA damage and stalled replication fork. The replication protein A (RPA) is a heterotrimeric complex consisting of 3 subunits - RPA1, RPA2, and RPA3. Upon genotoxic stress RPA is known to coat ssDNA and is subsequently hyperphosphorylated, initiating downstream DNA-damage response (DDR) pathways[25]. All three RPA subunits showed a thermal stabilization which likely reflect the ssDNA-bound form of RPA and hyperphosphorylation. Additionally, we observed a thermal destabilization of CHEK1, which was concomitant with its phosphorylation at Ser345 (Supplementary Fig. 2A) and further validates the initiation of DDR pathways. The fact that the thermal shifts are present early in sensitive and resistant cell lines suggest that resistance mechanisms occur downstream of these initial responses. DNMT1 is a major enzyme involved in DNA methylation inheritance and plays a critical role in maintaining genome stability[26]. Notably, similar to the FDA-approved DNMT1 inhibitor decitabine, gemcitabine is also a cytidine analog. We therefore investigated the possibility of a direct soluble gemcitabine triphosphate-DNMT1 interaction in a cell lysate western blot CETSA experiment but did not see significant thermal stability shifts (Supplementary Fig. 2B). Additionally,

gemcitabine treatment did not result in DNMT1 degradation as is described for decitabine (Supplementary Fig. 2C). It instead appeared plausible that the observed CETSA effects report on the modulations of specific protein or DNA interactions with DNMT1 induced at the stalled replication fork.

## Activation of apoptosis in sensitive cells versus checkpoint release for cell cycle progression in resistant cells

Through the use and analyses of several apoptosis-inducing drugs, we recently identified a prototypic CETSA apoptosis response that is dominated by nuclear proteins and reflects very early apoptosis including caspase activation[22]. This response was characterized by 47 proteins which we termed the core CETSA apoptosis ensemble (CCAE), and this provided the first means for direct assessment of caspase activation in intact cells. When our current data was compared with the ensemble described above, 37 and 34 proteins were measured for the sensitive and resistant cells, respectively (Supplementary Fig. 3). Among those, 23 proteins were identified as hits in sensitive, but only 1 in resistant cells (Fig. 3A). These results unequivocally conclude that apoptosis induction was indeed unique to sensitive cells, despite the far higher gemcitabine concentration used to treat resistant cells. To further verify that apoptosis is only induced in sensitive cells, we looked at PARP1 cleavage, a recognized hallmark of apoptosis, by

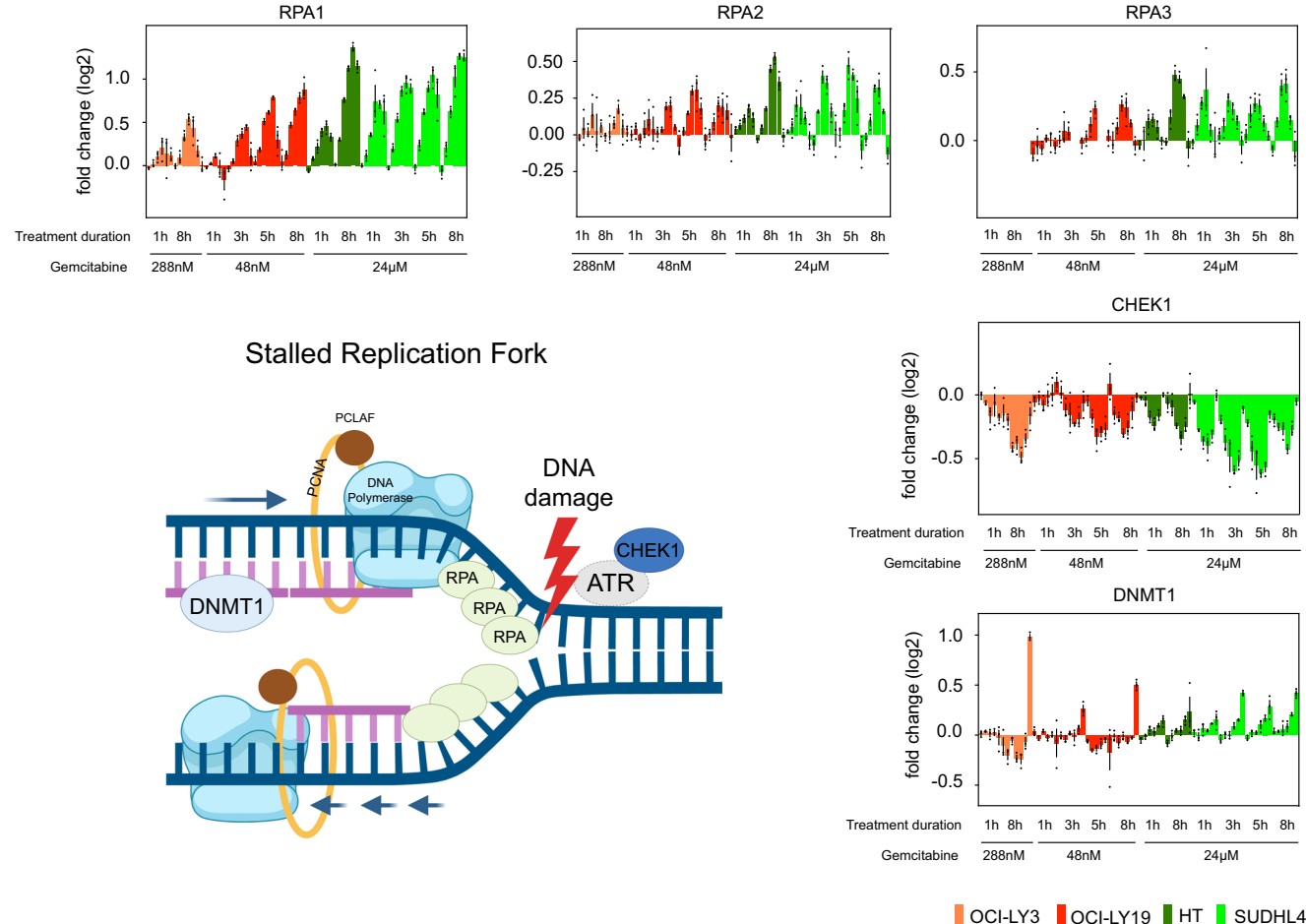

**Fig. 2 | Gemcitabine-induced stalled replication fork.** Hypothetical model indicating proteins at the stalled replication fork after gemcitabine-induced DNA damage, and respective IMRPINTS profiles of RPA1, RPA2, RPA3, CHEK1 and DNMT1 in sensitive OCI-LY3 (orange) and OCI-LY19 (red), as well as resistant HT (dark green) and SUDHL4 cells (light green) after 1 h, 3 h, 5 h or 8 h of gemcitabine treatment. Data are presented as mean log2 fold change compared to the reference ±SEM from biological replicates (*n* = 3). Source data are provided as a Source Data file. Created in BioRender. Tam, W. (2025) https://BioRender.com/6fy4aw8.

western blot. Indeed, we only observed cleaved PARP1 in sensitive cells (Supplementary Fig. 4A). Although apoptosis can be initiated by ATR/CHEK1 signaling via p53-activation[27], the lack of CETSA shifts in proteins recently defined as p53-regulated proteins in cell death, indicates that the observed processes here are independent of p53[28]. Our current study furthermore resolved the sequence of early apoptosis events and showed that the emergence of the CETSA apoptosis response in sensitive cells was clearly time dependent with proteins such as PARP1, XRCC5, XRCC6, MATR3, LMNB1, LMNB2, RBMX and ZC3H11A showing distinct thermal stability shifts as early as 3 h and 5 h following gemcitabine exposure (Fig. 3B). For some proteins that are cleaved by caspases, we also described a "regional stabilization due to proteolysis" (RESP) effect, whereby stability changes in regions either N- or C-terminal of caspase cleavage sites were observed. This effect was also measured here in a subset of proteins including known caspase targets such as PARP1, LMNB1, MATR3 and DDX21 (Fig. 3C).

In contrast to the prominent CETSA apoptosis signatures featured in the sensitive cells, we observed cell-cycle regulating processes as one of the dominant features in the response of resistant cells. Contributing to this were significant shifts for cyclins and cyclin-dependent kinases (CDKs), most prominently CCNA2, CCNB1, CCNB2 and CDK1 (Fig. 3D). These proteins showed distinct time-dependent thermal stabilizations or abundance changes with similar IMPRINTS profiles as compared to our previously published cell cycle study[23]. The shifts indicated increased activation of CDK complexes that promoted G2/M and G1/S phase checkpoint transitions. To rule out that these effects

are due to the higher gemcitabine concentration used in treatment of resistant cells, we consulted our additional low dose datasets that also included the same concentration as used for the sensitive cells (48 nM). The findings supported similar modulations of cell cycle checkpoints in resistant cells at lower doses, demonstrating that this induced response spanned over a wide concentration range (Supplementary Fig. 4B). Additionally, in our previous cell cycle study, RB1 phosphorylation during G1/S checkpoint release resulted in a thermal stabilization. Here, we observed the opposite effect, i.e. thermal destabilization and thus dephosphorylation, in gemcitabine sensitive cells (Supplementary Fig. 4C). Analysis of cell cycle distribution using propidium iodide staining confirmed G1 arrest in sensitive cells, while resistant cells underwent normal cycling upon gemcitabine treatment (Fig. 3E).

## DDR initiates translesion DNA synthesis as a resistance mechanism

Next, we sought to explain how resistant cells were able to proceed with the cell cycle, despite the exposure to a DNA synthesis inhibitor.

DDR is dependent on the availability of dNTPs at appropriate levels for accurate DNA synthesis. SAMHD1, which exhibited significant destabilization in resistant cells (Supplementary Fig. 5A), is a regulator of dNTP homeostasis via its dNTPase activity[29]. Like CHEK1, we tested whether phosphorylation of SAMHD1 is concomitant with its thermal destabilization, but did not detect significant changes (Supplementary Fig. 5B). When we knocked down SAMHD1 (Supplementary Fig. 5C) as

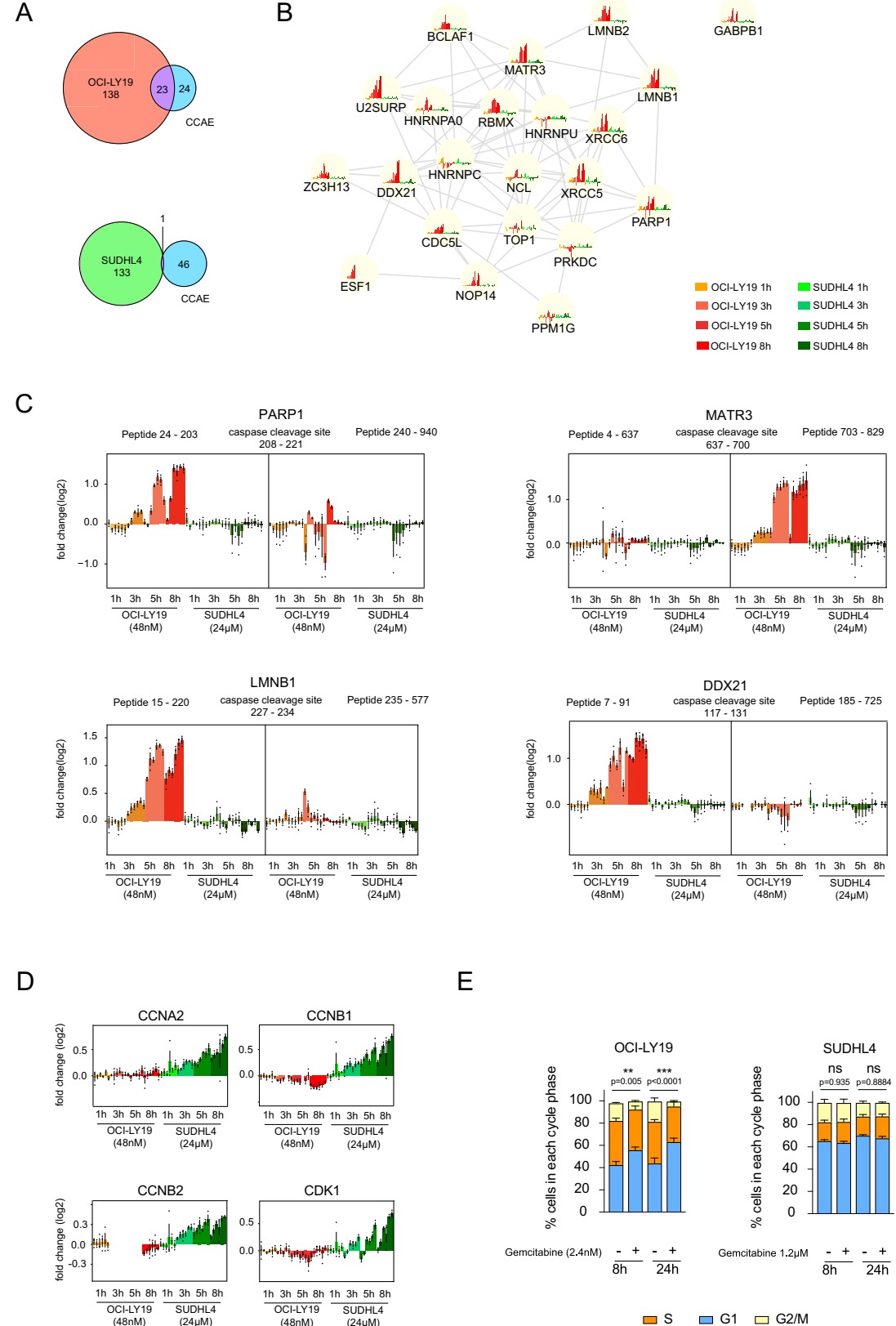

an attempt to re-establish gemcitabine sensitivity, we instead observed a slight increase in resistance (Supplementary Fig. 5D). LC/MS measurements of deoxyribonucleotide (and ribonucleotide) pools showed differences between SUDHL4 SAMHD1 WT and KO cells primarily in dGNP and dANP abundance; this is consistent with SAMHD1 being a dNTP hydrolase of purine nucleotides (Supplementary Fig. 5E). In a

more compact 3-temperature IMPRINTS-CETSA experiment we found similar gemcitabine responses between SUDHL4 WT and KO cells. However, a notable difference was a much weaker stabilization of RRM1 after gemcitabine treatment in the KO cells (Supplementary Fig. 5F). As gemcitabine-triphosphate is a substrate of SAMHD1[30,31], we reasoned that the reduced cycling between different

**Fig. 3 | CETSA responses in gemcitabine sensitive versus resistant cells. A** Venn diagram showing overlap of the hits from sensitive OCI-LY19 (top) and resistant SUDHL4 cells (bottom) with the previously identified CCAE (Core CETSA Apoptosis Ensemble) proteins. **B** STRING plot showing the overlapping proteins from A with IMPRINTS CETSA profiles for OCI-LY19 (red hues) and SUDHL4 (green hues) after 1 h, 3 h, 5 h and 8 h gemcitabine treatment. Data are presented as mean log2 fold change compared to the reference from biological replicates (*n* = 3). Source data are provided as a Source Data file. **C** IMPRINTS profiles of PARP1, MATR3, LMNB1 and DDX21 showing peptides before and after known caspase cleavage sites in sensitive OCI-LY19 cells (red hues) and resistant SUDHL4 cells (green hues) after 1 h, 3 h, 5 h and 8 h gemcitabine treatment. Data are presented as mean log2 fold

change compared to the reference ±SEM from biological replicates (*n* = 3). Source data are provided as a Source Data file. **D** IMPRINTS profiles of CCNA2, CCNB1, CCNB2 and CDK1 for OCI-LY19 (red hues) and SUDHL4 (green hues) after 1 h, 3 h, 5 h and 8 h gemcitabine treatment. Data are presented as mean log2 fold change compared to the reference ±SEM from biological replicates (*n* = 3). Source data are provided as a Source Data file. **E** Progression of cell cycle and distribution of cells in different cell cycle phases in the sensitive OCI-LY19 and resistant SUDHL4 cells after 8 h and 24 h with and without gemcitabine treatment. A two-way ANOVA was performed, and data are presented as relative percentage of cells in each cycle phase ±SEM from biological replicates (*n* = 4) with *p*-values denoting significant changes in G1 phase (blue). Source data are provided as a Source Data file.

phosphorylation states of gemcitabine in the SAMHD1 KO cells affected the cellular concentration of the inhibitory diphosphate form of gemcitabine, thereby causing the reduction of RRM1 engagement. This might subsequently lead to the observed attenuated deoxyribonucleotide pools in the KO cells, explaining the increase in resistance.

Interestingly, "translesion synthesis" (TLS) appeared as a prominent pathway only in resistant cells at 8 h (Fig. 4A). TLS is a process that facilitates DNA synthesis over damaged lesions by reorganizing replication complexes through the recruitment of specialized DNA repair polymerases. In addition to subunits of ssDNA binding proteins RPA1, RPA2 and RPA3, we observed pronounced time dependent thermal stabilization and abundance increases of two key proteins associated with TLS: PCNA binding protein (PCLAF) and Denticleless Protein Homolog (DTL). Accompanying these changes, we observed strong thermal destabilization of the core catalytic subunits of the replicative DNA polymerase δ (PolD), POLD1, POLD2 and POLD4, the latter also depicting a decrease in abundance levels. To investigate the possible induction of dedicated TLS polymerases as a putative mechanism to overcome DNA damage and hence gemcitabine resistance, we examined the protein levels of several of the repair/TLS polymerases after 1 h, 3 h, 5 h and 8 h of gemcitabine exposure in both sensitive and resistant cells. POLη, POLι, and Rev1 did not show any difference in protein levels (Supplementary Fig. 6A). Notably, however, we observed a time-dependent reduction of POLκ protein abundance only in sensitive cells (Supplementary Fig. 6B), which was restored in the presence of the pan-caspase inhibitor zVAD-FMK (Supplementary Fig. 6C). This suggests an active elimination of DDR mechanisms through caspases upon the irreversible commitment to apoptosis. To further validate that the observed CETSA shifts indeed reported on TLS activation, we employed an orthogonal standard TLS assay, whereby the ubiquitination status of the DNA clamp protein, PCNA, is assessed. Upon DNA damage, mono-ubiquitination of PCNA primes access to DNA by TLS polymerases[32]. We measured the levels of total versus mono-ubiquitinated PCNA after gemcitabine treatment and indeed only observed TLS activation in resistant cells (Fig. 4B). Next, we sought to explore the effect on gemcitabine resistance by disrupting TLS with a REV7/REV3 interaction inhibitor. We indeed found synergistic effects between gemcitabine and REV7/REV3-In-1 (Fig. 4C). From these findings, we postulate a gemcitabine resistance mechanism that involves the release of replicative DNA polymerase (PolD), indicated by thermal destabilizations, followed by mono-ubiquitination of PCLAF, which facilitates access to TLS polymerases and restart of replication fork to bypass DNA damage-induced replication arrest and apoptosis (Fig. 4D).

**Auxiliary DNA damage repair (ADDR) response: a protein ensemble induced by genotoxic drugs to drive resistance**

In addition to the TLS CETSA protein shifts, an ensemble of 5 proteins exhibited a strong concomitant protein abundance increase following gemcitabine treatment. This ensemble includes: RRM2 (Ribonucleoside diphosphate Reductase subunit M2) and TK1 (Thymidylate Kinase), which are involved in deoxyribonucleotide provision; GMNN (Geminin), which inhibits the formation of a pre-replication complex;

SLBP (Stem Loop Binding Protein), which promotes histone transcription, and FBXO5 (F-box only protein 5), a regulator of the anaphase promoting complex. Together with DTL and PCLAF, these proteins were distinctly upregulated in the two resistant, but not sensitive DLBCL cell lines. We termed this protein ensemble, the Auxiliary DNA Damage Repair (ADDR) response proteins (Fig. 5A). We confirm that the ADDR CETSA signature was also present with lower doses of gemcitabine exposures in resistant cells (Supplementary Fig. 7A). We next examined whether the ADDR ensemble is more commonly activated in these cells and indeed found increased abundances upon treatment with other DNA damaging drugs such as cladribine and cytarabine (Fig. 5B). To further validate whether the ADDR response is a conserved mechanism, we utilized a completely different cell system, MDA-MB-231 breast cancer cells, treated with another class of genotoxic drug, the DNA cross-linking agent cisplatin. Strikingly, in the dataset of this model, all 5 proteins of the ADDR response, as well as DTL and PCLAF, were among the strongest shifting proteins which displayed abundance changes (Supplementary Fig. 7B). These observations indicate that the ADDR program may have a broader role in conferring resistance towards a spectrum of DNA damaging agents. Notably, prominent thermal destabilization of PolD subunits (Supplementary Fig. 7C), described above, were also observed in the cisplatin data. However, in contrast to gemcitabine-treated DLBCL cells, there was no shift for proteins in the ssDNA binding RPA complex or CHEK1 (Supplementary Fig. 7D) in cisplatin-treated breast cancer cells. Hence, across both experimental models, our data pointed to a CHEK1-independent mechanism for inducing TLS and ADDR responses. Finally, we confirmed that the gemcitabine-resistant SUDHL4 cells were, in fact, also resistant towards cytarabine, cladribine as well as cisplatin (Fig. 5C).

**ATR inhibition abrogates gemcitabine resistance through attenuation of TLS and ADDR induction**

Given the early CETSA signals of DNA damage sensing proteins upon gemcitabine treatment, and the role of TLS and ADDR responses in resistant cells, we reasoned that preventing the initiation of DDR mechanisms might be exploited to re-establish sensitivity. A key player in sensing DNA damage, together with RPA and CHEK1, is the serine/threonine kinase ATR. The use of inhibitors of ATR kinase (ATRi) have shown pre-clinical and clinical synergy with gemcitabine[33]. Accordingly, we investigated the effects of the ATRi, AZD6738, on the gemcitabine response in resistant SUDHL4 cells. Interestingly, combination treatment resulted in attenuation of gemcitabine resistance as observed by a 100 fold lower $IC_{50}$ value at 72 h (Fig. 5D).

Next, we investigated whether the resistance signatures in TLS and ADDR responses are affected and thus performed a 3-temperature IMPRINTS experiment in resistant SUDHL4 cells treated with either gemcitabine alone, AZD6738 alone, or in combination. Consistent with our hypothesis, the most prominent effects were seen for the proteins of the ADDR ensemble, as well as DTL and PCLAF from the TLS ensemble, whereby there was a dramatic decrease in abundance (Fig. 5E). The destabilisation of POLD1 and POLD2, and level changes of POLD4, were also attenuated by ATRi (Fig. 5F). This strongly supported

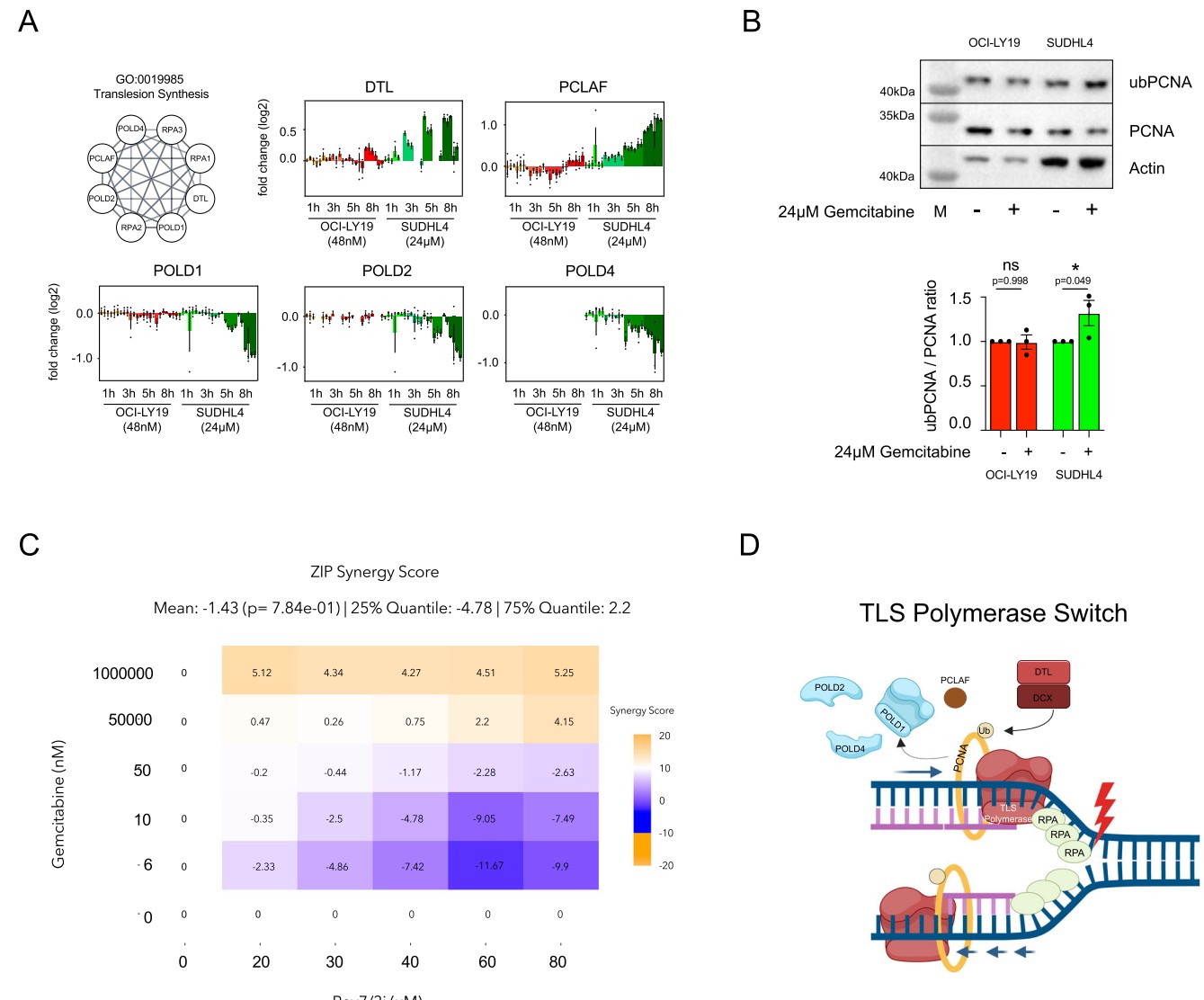

**Fig. 4 | Translesion synthesis in gemcitabine resistant cells. A** Nodes indicating translesion synthesis pathway as a GO term and IMPRINTS profiles of the involved proteins in OCI-LY19 (red hues) and SUDHL4 (green hues) after 1 h, 3 h, 5 h and 8 h gemcitabine treatment. Data are presented as mean log2 fold change compared to the reference ±SEM from biological replicates (*n* = 3). Source data are provided as a Source Data file. **B** Representative western blot of PCNA and mono-ubiquitinated PCNA (top) and quantification of ubPCNA/PCNA ratio (bottom) in the sensitive OCI-LY19 (red) and resistant SUDHL4 (green) cells after 6 h of gemcitabine (or vehicle) treatment. A two-way ANOVA was performed comparing vehicle versus gemcitabine treatment, and data are presented as mean ubPCNA to total PCNA ratio ±SEM from biological replicates (*n* = 3). Source data are provided as a Source Data file. **C** ZIP Synergy score of gemcitabine and rev7/3-in-1 concentrations in SUDHL4 cells at 48 h. **D** Hypothetical model of gemcitabine induced translesion synthesis polymerase switch. Created in BioRender. Tam, W. (2025) https://BioRender.com/0i85r9z.

the notion that the induction of these proteins was indeed ATR-dependent. The re-established sensitivity by ATRi reinforced the conclusion that the induction of the ADDR and TLS (DTL/PCLAF) ensembles was a key prerequisite for establishing resistance to DNA-damaging drugs.

The quite dramatically decreased abundance of the ADDR ensemble upon combination treatment could be attributed to the relatively fast turnover rates of these proteins in exponentially growing cells, typically accomplished by a high rate of production and degradation. Indeed, in a Molm16 AML cell line protein turnover dataset used in our lab as reference, the 4 measured proteins all have rapid turnover rates (TK1-15 h; RRM2-11h; SLBP-28h; PCLAF-10h) (Supplementary Data 2). Arguably, this design makes these proteins particularly useful for regulating urgent events in cellular processes. However, our current data is not conclusive on whether this is only due to decreased transcriptional activity for the corresponding genes in ATRi

treated resistant cells, or whether there are posttranscriptional mechanisms or activation of proteosome degradation components induced. In the future, a more detailed elucidation of the signaling mechanisms post-ATR will be helpful to define contributions from different mechanisms to increased protein levels.

## CETSA signature responses in DLBCL clinical samples
To test the feasibility of applying MS-CETSA in clinical samples, we performed an ITDR-CETSA experiment on biopsies from two DLBCL patients who have relapsed after first line therapy and have not been previously treated with gemcitabine. Cells were extracted using Ficoll-paque and treated ex vivo for 5 h with increasing doses of gemcitabine (Fig. 6A). By comparing the CETSA signatures of resistant and responding cells, we can conclude that both clinical samples were dominated by shifts in proteins of the CETSA apoptosis ensemble discussed above (Fig. 6B), i.e. still sensitive to gemcitabine.

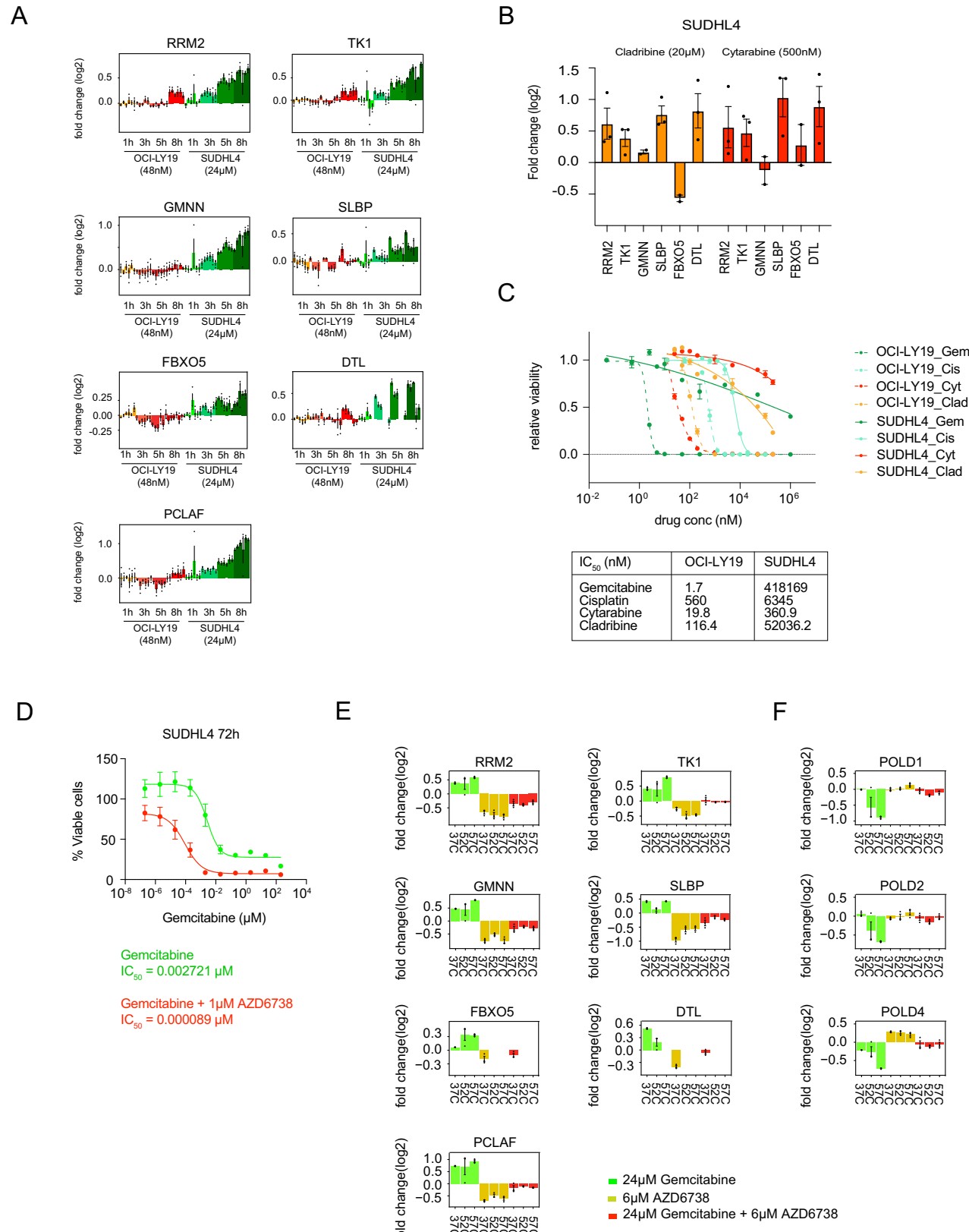

**Static omics studies to identify drug resistance mechanisms**
Using a time-resolved IMPRINTS CETSA approach we identify a potential candidate drug-induced resistance mechanism in DLBCL cells and show that the responses in sensitive and resistant cells were largely mutually exclusive (Supplementary Fig. 8A, B).

When most omics studies of drug resistance are performed through profiling cells in the absence of treatment, we wondered whether the stark divergence seen in the late time points of our gemcitabine-induced CETSA data could be captured through such static omics approaches, and if similar informative conclusions could

**Fig. 5 | ADDR response in gemcitabine resistant cells. A** IMPRINTS profiles of ADDR protein ensemble in OCI-LY19 (red hues) and SUDHL4 (green hues) after 1 h, 3 h, 5 h and 8 h gemcitabine treatment. Data are presented as mean log2 fold change compared to the reference ±SEM from biological replicates (*n* = 3). Source data are provided as a Source Data file. **B** Quantification of ADDR proteins in SUDHL4 cells after 6 h treatment with cladribine (orange) and cytarabine (red). Data are presented as mean log2 fold change compared to the reference ±SEM from biological replicates (*n* = 3). Source data are provided as a Source Data file. **C** Relative viability and IC$_{50}$ values of OCI-LY19 (dotted lines) and SUDHL4 (continuous lines) cells after 48 h treatment with increasing concentrations of gemcitabine (green), cisplatin (light blue), cytarabine (red) or cladribine (orange). Data are presented as mean relative viability compared to the reference ±SEM from biological replicates (*n* = 3). Source data are provided as a Source Data file.

**D** Relative viability of SUDHL4 cells treated for 72 h with increasing concentrations of gemcitabine, either alone (green) or in combination with 1 μM AZD6738 (red). Data are presented as mean relative viability compared to the reference ±SEM from biological replicates (*n* = 9). Source data are provided as a Source Data file. **E** IMPRINTS profiles of ADDR protein ensemble in gemcitabine resistant SUDHL4 cells after 6 h of treatment of gemcitabine alone, AZD6738 alone or in combination. Data are presented as mean log2 fold change compared to the reference ±SEM from biological replicates (*n* = 3). Source data are provided as a Source Data file. **F** IMPRINTS profiles of POLD1, POLD2, POLD4 in gemcitabine resistant SUDHL4 cells after 6 h of treatment of gemcitabine alone, AZD6738 alone or in combination. Data are presented as mean log2 fold change compared to the reference ±SEM from biological replicates (*n* = 3). Source data are provided as a Source Data file.

be made on candidate resistance mechanisms. Therefore, we investigated sequence data of the cell lines used from the "CCLE Cell Line Gene Mutation Profiles" database to examine the genetic difference of our cell lines. This shows similar number of mutations in all cell lines except the resistant HT cells (OCI-LY3: 90, OCI-LY19: 77, SUDHL4: 79, HT: 256), which had a significantly higher number of mutations that seem to be cell line specific (Supplementary Fig. 8C). An over-representation analysis (Supplementary Fig. 8D) did not reveal any pathways that are specifically affected in sensitive cells. For resistant cells we noted the term "intrinsic apoptotic signaling pathway in response to DNA damage by p53 class mediator" enriched, which could be explained by the mutational status of p53 for these cells. However, according to our data, induction of apoptosis in sensitive DLBCL cells is p53 independent.

Furthermore, we profiled the four cell lines on the protein level by quantitative proteomics. In a correlation heatmap we show that the 2 sensitive and resistant cell lines cluster together, respectively, indicating baseline differences that may already reflect their drug response phenotypes (Supplementary Fig. 8E). We specifically compared the resistant SUDHL4 with the sensitive OCI-LY19 cells and detected 111 proteins down regulated, and 150 proteins up regulated. Notably, no difference was observed in most of the above-described proteins related to gemcitabine resistance (Supplementary Fig. 8F) and again, no specific pathways indicating drug resistance mechanisms were overrepresented (Supplementary Fig. 8G, H). Instead, enriched GO terms were linked to general B-cell functions (e.g. receptor signaling, cytokine production) and we therefore concluded that these were the baseline differences reflected in the clustering of sensitive and resistant cells.

Taken together, this analysis of genetic mutations and a quantitative proteomic analysis support that baseline profiling of cell lines is, at least in this case, insufficient to predict gemcitabine resistance. Therefore, global MS-CETSA time-resolved characterization of resistance mechanisms will likely in many cases constitute a more informative approach for identifying cancer drug resistance mechanisms.

## Discussion

Non-hypothesis driven system-wide methods have the potential to identify the most prominent molecular processes regulating cellular phenotypes. However, despite cellular biochemistry controlling most molecular processes of the cell, methods for efficient studies of cellular biochemistry at the systems level have been elusive. This has, arguably, also contributed to our relatively fragmented current understanding of the biochemical basis for pathway activation in cancer drug resistance.

CETSA constitutes the first systems-wide method which can report on a range of different types of cellular biochemistry, from protein-protein and protein-DNA/RNA interactions to phosphorylation events and flux through metabolic pathways[21]. However, so far CETSA studies have predominantly been focused on identifying drug interactions. Although it has been clear that CETSA can report on cellular

pathway modulations downstream of drug binding, in our view this has not yet been systematically explored. Limitations of previous approaches have been the use of suboptimal CETSA implementations which don't allow for robust measurements of small stability shifts typically induced by functional biochemical changes. Furthermore, time-dependent studies have not been systematically explored to dissect sequences of drug-induced activation of cellular processes/pathways. In one study, we have previously applied a 2 time-point MS-CETSA approach to study MoA and resistance to 5-FU, which revealed attenuation of anticipated toxic biochemistry in resistant cells, but no drug-induced resistance response[34].

In the present work we use the highly sensitive IMPRINTS-CETSA implementation in a time-dependent approach to demonstrate applicability of this technology to study the biochemical pathways involved in gemcitabine MoA and resistance mechanisms. By focusing on the overlapping responses in cell pairs of resistant and sensitive cells, a distinct view of the biochemistry of the MoA of gemcitabine in the two cell types is revealed. The initial responses are very similar, reflecting the direct target engagement of RNR, through RRM1 thermal stabilization, as well as the establishment of a DNA-damage signaling hub activated by RPA binding to exposed ssDNA and CHEK1 phosphorylation. The almost identical isothermal dose response CETSA shifts for RRM1 indicate an exclusion of modifications in drug internalization and metabolism as dominant resistant mechanisms in our system. At the 3–8 h time-points, sensitive cells rapidly enter apoptosis as judged from the observed shifts in CETSA apoptosis ensemble proteins, while resistant cells show CETSA shifts of CDK complexes supporting open cell cycle checkpoints, consistent with the continued proliferation.

Most notably in the resistant cell CETSA data, distinct responses are seen for proteins related to activation of DNA repair, i.e., the abundance and thermal stability changes of two TLS biomarkers (PCLAF and DTL) as well as prominent destabilization in Polδ, likely reporting on the induction of TLS. This is further supported by the increased mono-ubiquitination of PCNA only in the resistant cells and the synergistic effects of gemcitabine with REV7/REV3 interaction inhibitor. The induced TLS program explains how resistant cells overcome stalled replication forks by allowing DNA-synthesis over damage lesions. TLS has not been previously implied in resistance to gemcitabine but has been suggested as a mechanism of resistance to cisplatin as derived from over-expression of TLS polymerases in resistant cells[35]. However, in these cases TLS proteins are assumed to be constitutively expressed and not part of an induced TLS response, as uncovered in the present study.

In addition to the induction of CDK activation and TLS programs, the induction of the ADDR ensemble of proteins is the most dominating feature of the response in resistant cells. These proteins appear to have functions that can support DNA-repair/replication and could therefore be supportive for TLS, although not previously identified as an ensemble in a DNA repair context. The distinct attenuation of both the induction of TLS proteins DTL and PCLAF, and the ADDR response

A

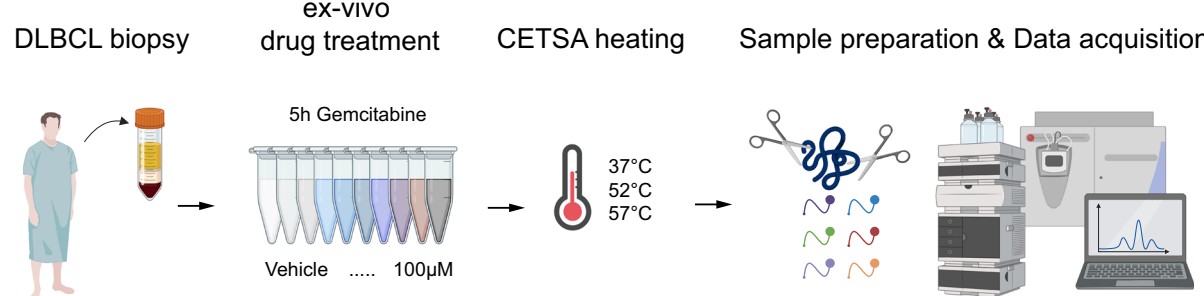

B

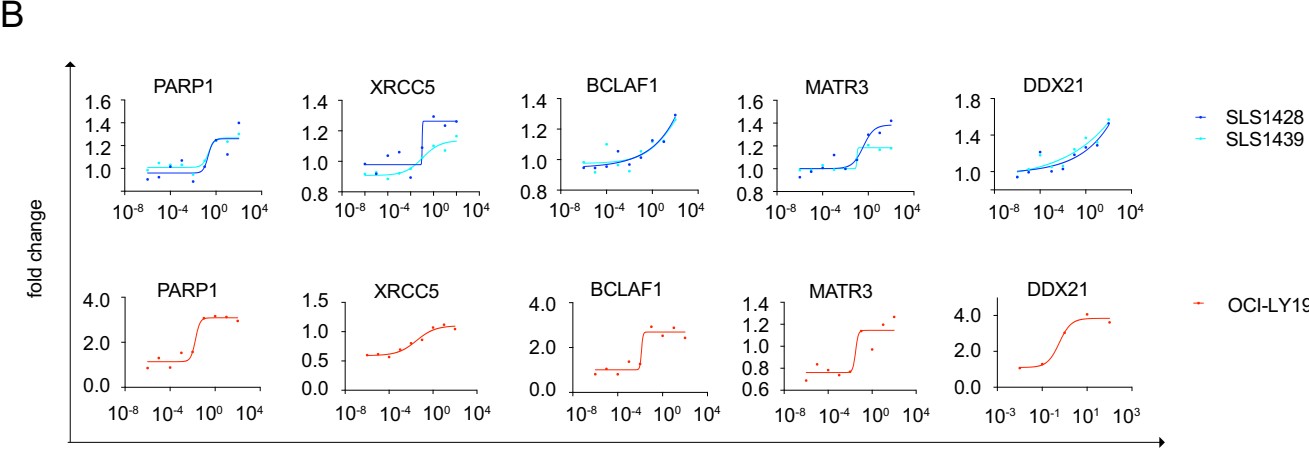

**Fig. 6 | Stratification of clinical samples into sensitive and resistant type using MS-CETSA. A** Experimental design for MS-CETSA treatment of patient samples from DLBCL patients. Patient samples were treated with different doses of gemcitabine ex vivo for 5 h. The treated samples were CETSA heated and subjected to mass spectrometry. Created in BioRender. Tam, W. (2025) https://BioRender.com/ 4myrn1e. **B** MS-CETSA ITDR profiles of selected CCAE proteins in patient samples (top panel) and sensitive OCI-LY19 (bottom panel) cells. Data are presented as mean log2 fold change compared to the reference from technical replicates (n = 2). Source data are provided as a Source Data file.

by an ATR inhibitor strongly support that ATR is a signaling node in this response. However, we conclude that the response is likely not CHEK1 dependent, when ADDR response for cisplatin in MDA-MB-231 breast cancer cells does not coincide with CHEK1 activation. The disparity likely reflected differences in DNA damage mechanisms between the two drugs: cisplatin is a DNA crosslinking agent while gemcitabine induces single strand breaks.

Intriguingly, CHEK1 activation is expected to mediate cell cycle arrest, but in contrast, in gemcitabine resistant cells, the CETSA shifts of CDK complex and cell cycle assessments support the opposite effect, i.e., opening of cell cycle checkpoints. This gives further support for the activation of an alternative signaling pathway for the induction of a pathway downstream of ATR, controlling DNA-repair and cell cycle checkpoints to support cell proliferation during genotoxic challenges, principle of this pathway outlined in Fig. 7. However, despite significant efforts we have not been able to identify additional components of the signaling pathway downstream of ATR, which also might provide additional target proteins for specifically attenuating gemcitabine resistance.

In addition to constituting a pathway for induction of DNA-repair, the fact that ATR inhibition re-sensitized cells to gemcitabine, supports that this response is a key component of the gemcitabine resistance in this system. There have been previous reports of positive results for

using ATRi in combination with gemcitabine in pancreatic cancer[33] and ovarian cancer[36] therapies. In a recent phase 2 trial in platinum-resistant high-grade serous ovarian cancer, a combination of the selective ATR inhibitor berzosertib, and gemcitabine showed significantly prolonged progression-free survival compared to treatment with gemcitabine alone[37]. The current studies provide a mechanistic rationale for the combination of ATRi and gemcitabine for DLBCL.

Notably, this study emphasizes the importance of monitoring drug-induced responses as an approach to successfully identify resistance mechanisms as analyses of genetic mutations and static proteomic profiling failed to capture the proposed gemcitabine resistance mechanisms. As a future strategy for patient stratification, CETSA could potentially be used to monitor whether this (or other) resistance mechanism(s) is in effect in clinical samples, or if instead the early apoptosis profile are detectable with CETSA, indicating sensitivity. The data from two clinical DLBCL patient samples of gemcitabine naïve patients also support that high quality CETSA information can be obtained from clinical DLBCL samples. In spite of cell heterogeneity in typical tumor samples, we envisage that CETSA can identify the dominant resistance mechanism(s) in the sample, to guide therapy, and as resistance evolves anew, again dominant resistance mechanisms might be detectable by CETSA.

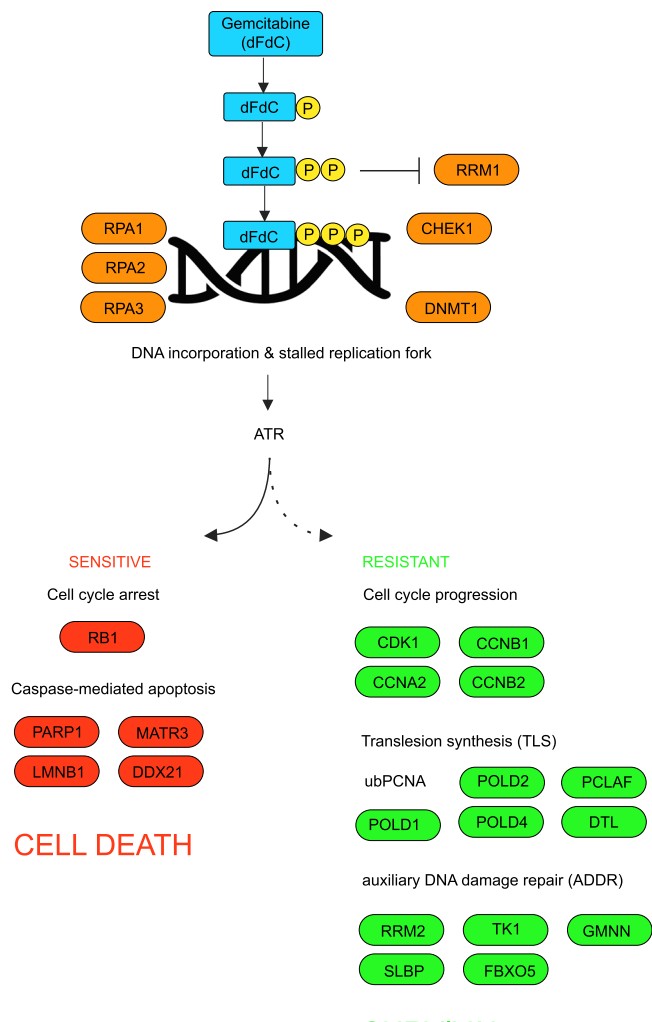

**Fig. 7 | Gemcitabine-induced cellular responses.** Schematic summary of gemcitabine-induced cellular responses in sensitive (red nodes) versus resistant (green nodes), or both (orange nodes) cells. Dotted line indicates signaling pathway yet to be established in detail. Created in BioRender. Tam, W. (2025) https://BioRender.com/jgju7ac.

Together the present study supports that time-dependent IMPRINTS-CETSA constitutes a highly efficient strategy to discover sequences of prominent pathway activation explaining cancer drug MoA and resistance. Therefore, as an alternative to focused studies of cancer drug MoA, which are often limited in their scope by requirement of pathway/protein specific biochemical assays, this work establishes IMPRINTS-CETSA as an efficient strategy for global studies of the biochemistry of cancer drug resistance, where comprehensive insights into effects on many different cellular pathways can be directly accessed using a single method. These studies also provide a repertoire of MoA-based drug resistance biomarkers showing robust responses with potential applicability in the clinic.

## Methods

This research complies with all relevant ethical regulations. Collection of patient samples is approved under A*STAR IRB: 2021-140.

### Resource availability

Key resource are provided as Supplementary table 1. Further information and requests for resources and reagents should be directed to and will be fulfilled by the corresponding author.

## Experimental model and subject details

**Cell lines.** Human breast adenocarcinoma cell line MDA-MB-231 (CRM-HTB-26) was purchased from ATCC. Human lymphoma cell line SUDHL4 (CRL-2957) was purchased from ATCC, HT cells (CRL-2260) were a gift from the lab of Ernesto Guccione, Icahn School of medicine at Mt Sinai (formerly at IMCB, Singapore), OCI-LY19 and OCI-LY3 was obtained from the lab of Manikandan Lakshmanan at IMCB, Singapore.

All the DLBCL cell lines and MDA-MB-231 were maintained in RPMI-1640 medium (R8758, Sigma) and L-glutamine, supplemented with 20% fetal bovine serum (FBS), 100 units/ml penicillin and streptomycin in a 37 °C CO2 incubator.

**Generation of SUDHL4 SAMHD1 knockout cells.** Knockout was performed using LentiCRISPRv2GFP vector (82416, Addgene). Single-guide RNA encoding SAMHD1 was cloned into LentiCRISPRv2GFP vector. Briefly, lentiviruses were packaged using HEK293T cells via co-transfection of gene of interest, VSVG and delta 8.2 vector, using Lipofectamine 2000 transfection reagent (11668019, Thermo Fisher). Viruses collected were concentrated using Amicon Ultra Centrifugal filters (C2566709, Merck) and spinoculated onto the SUDHL4 cells in the presence of 8 μg/ml polybrene (sc-134220, Santa Cruz) at 800 g for 30 min at room temperature. The target sequences of the sgRNAs are as follow: SAMHD1 sgRNA-1 forward 5'-CACCGAGGATGTCTAGTTC ACGCAC -3'; SAMHD1 sgRNA-1 reverse 5'-AAACGTGCGTGAACTAGAC ATCCTC -3'. Single cells containing the CRISPR-GFP positive vector were then sorted through FACS and harvested as monoclones.

**Primary DLBCL clinical patient samples.** We have complied with all relevant ethical regulations. Collection of samples is approved under A*STAR IRB: 2021-140. Informed written consent was obtained from all participants and no compensation was provided. Tumors were collected in MACS Tissue Storage Solution (130-100-08, Miltenyi) and kept on ice for transport. Tissue was cut into equally small pieces using a scalpel. To obtain single cell solutions, the cells were passed through a sterile 70 μM cell filter mesh (352350, Corning) in RPMI-1640 medium (R8758, Sigma) and L-glutamine, supplemented with 10% fetal bovine serum (FBS), 100 units/ml penicillin and streptomycin. Cells number was determined and MS-CETSA experiment was performed immediately.

## Method details

**Selected drugs.** Gemcitabine, AZD6738 (kindly provided by Prof. Anand Jeyasekharan, CSI, Singapore) and Decitabine were solubilized in water. Cisplatin, Cytarabine, Cladribine, Z-VAD-FMK, MG132 and REV7/REV3L-IN-1 were solubilized in DMSO. All compound stocks were aliquoted and stored at −20 °C.

**MTT cell viability assay.** Cell viability was assessed using the MTT assay. Cells were seeded in 96-well v-bottom plates at a density of $2 \times 10^4$ cells per well and incubated overnight. After drug treatment, 100 μL of MTT solution (0.5 mg/mL in PBS) was added to each well and incubated at 37 °C for 2 h. The resulting formazan crystals were dissolved in DMSO, and absorbance was measured at 540 nm using a microplate reader. Cell viability was expressed as a percentage relative to untreated controls.

### The cellular thermal shift assay (CETSA)

**CETSA in intact cells.** For the in vitro IMPRINTS-CETSA experiments, cell lines were seeded at $0.5 \times 10^6$ cells/ml of media and preconditioned in complete RPMI with 2% FBS for 24 h. The cells were then treated with either vehicle or drug at their respective final concentrations and incubated at 37 °C and 5% $CO_2$ for indicated time points. Cells were pelleted for 4 min at $400 \times g$, washed with PBS and resuspended in 50 μl PBS. For the in vitro ITDR-CETSA experiments, cell lines were distributed into 6 tubes at $0.3 \times 10^6/100$ μl in media, while the total cells

of primary DLBCL clinical samples were distributed into 6 tubes in media. Cells were then treated with either vehicle or drug at their respective final concentrations and incubated at 37 °C and 5% CO$_2$ for indicated time points. Cells were pelleted for 4 min at 400 ×$g$, washed with PBS and resuspended in 50 μl PBS. Harvested cells and lysates were aliquoted into PCR tubes corresponding to each treatment condition and subjected to a 3 min CETSA heating step in a Veriti thermal cycler (Applied Biosystems) with temperatures ranging from 37 °C to 57 °C, followed by 3 min cooling at 4 °C.

**Cetsa in cell lysates.** For lysate CETSA experiments 20 ×10$^6$ cells/ml were lysed by adding 2X kinase buffer to the final concentration of 50 mM HEPES pH 7.5, 5 mM beta-glycerophosphate, 0.1 mM sodium orthovanadate (Na3VO4), 10 mM MgCl2, 1 mM TCEP (Sinopharm Chemical Reagent Co.), 1x protease inhibitor cocktail (Nacalai Tesque Inc.) and 25U/ml Benzonase. Cells were subjected to five freeze-thaw cycles with liquid nitrogen to release soluble proteins. The suspension was then centrifuged for 20 min at 20,000 ×$g$ and 4 °C to remove cell debris and 30 μl of supernatants were treated with either vehicle or drug at their respective final concentrations for 1 min. Lysates were aliquoted into PCR tubes corresponding to each treatment condition and subjected to a 3 min CETSA heating step in a Veriti thermal cycler (Applied Biosystems) with temperatures ranging from 37 °C to 57 °C, followed by 3 min cooling at 4 °C.

**Cell lysis and soluble protein extraction.** Following CETSA heat treatment, the cells were lysed by adding 2X kinase buffer to the final concentration of 50 mM HEPES pH 7.5, 5 mM beta-glycerophosphate, 0.1 mM sodium orthovanadate (Na3VO4), 10 mM MgCl2, 1 mM TCEP (Sinopharm Chemical Reagent Co.), 1x protease inhibitor cocktail (Nacalai Tesque Inc.) and 25U/ml Benzonase. All the samples were subjected to five freeze-thaw cycles with liquid nitrogen to release soluble proteins. For lysate CETSA experiments this step was skipped and immediately proceeded to the next step. The suspension was then centrifuged for 20 min at 20,000 ×$g$ and 4 °C to remove cell debris. The supernatants were then analyzed using either LC-MS or western blotting.

For quantitative proteomic profiling, cells were lysed in 8 M Urea with 1:250 protease inhibitor for 10 min at room temperature, followed by 3x pulse sonication (8 W, 30% amplitude, 30 s on, 10 s off). cell suspension was centrifuged for 20 min at 20,000 ×$g$ and 4 °C to remove any remaining cell debris. The supernatants were used for LC-MS or western blotting.

**Cell cycle analysis and flow cytometry.** Cell lines were seeded at 0.5 × 10$^6$ cells/ml of media and preconditioned in complete RPMI with 2% FBS for 24 h. The cells were then treated with either vehicle or drug at their respective final concentrations and incubated at 37 °C and 5% CO$_2$ for indicated time points. Cells were pelleted for 4 min at 400 ×$g$, washed with PBS. Cells were fixed in 70% ethanol overnight, washed twice with cold PBS, then resuspended in PI staining solution (100 μg/ml ribonuclease A, 50 μg/ml PI in PBS) and incubated in the dark for at least 30 min at room temperature, followed by flow cytometric analysis on a LSR II (BD Biosciences, UK) flow cytometer. FlowJo (FlowJo, LLC, USA) was used to analyze the data.

**Nucleotide quantification by LC-MRM/MS.** Cell lines were seeded at 0.5 × 10$^6$ cells/ml of media and preconditioned in complete RPMI with 2% FBS for 24 h. The cells were then treated with either vehicle or drug at their respective final concentrations and incubated at 37 °C and 5% CO$_2$ for indicated time points. Cells were pelleted for 4 min at 400 ×$g$, washed with PBS. Cell pellets were harvested and snap frozen in liquid nitrogen. The samples were stored in −80 °C until transportation to Creative Proteomics for nucleotide quantification through LC-MS analysis. Each cell sample was resuspended in 500 μl of 80% methanol

and then lysed on a MM 400 mill mixer at a shaking frequency of 30 Hz and with the aid of two metal balls for 2 min. The samples were subsequently sonicated for 1 min in an ice-water bath before centrifugal clarification at 21,000 ×$g$ and 5 °C for 10 min. The clear supernatants were collected for the following assay. The precipitated pellets were used for protein assay using a standardized BCA procedure. Serially diluted standard solutions of the targeted nucleotides were prepared in 80% methanol. 100 μl of each standard solution of the clear supernatant of each sample were dried under a nitrogen gas flow. The residues were dissolved in 100 μl of a 13C-labeled internal standard solution. 10 μl aliquots of the resulting solutions were injected into a C18 column (2.1 × 110 mm, 1.9 μm) to run UPLC-MRM/MS with (−) ion detection on a Waters Acquity UPLC system coupled to a Sciex QTRAP 6500 Plus MS instrument, with the use of tributylamine buffer (A) and acetonitrile (B) as the mobile phase for gradient elution.

**Western blot.** Western blotting was performed on protein extracts obtained either by freeze-thawing or lysis by RIPA buffer (Thermo Scientific). Protein concentrations for each sample were quantified using bicinchoninic acid (BCA) assay according to manufacturer's instructions.

Protein extract samples were mixed with NuPAGE loading buffer consisting of NuPAGE LDS sample buffer (NP0008, Life technologies) and reducing agent (NP0009, Life Technologies) and boiled at 95 °C. Proteins were separated on NuPAGE 4–12% Bis-Tris midi gels (WG1403BX10, Invitrogen) for 45–55 min at 200 V. Separated proteins were transferred to nitrocellulose membranes using the iBlot system (Invitrogen)onto nitrocellulose membranes. Membranes were blocked in 5% (w/v) non-fat milk (Semper AB) in TBS with 0.05% Tween 20 (Medicago 09-7510-100) (TBS-T) for 1 h with gentle shaking. Incubation with primary antibody was performed overnight at 4 °C and with gentle shaking. After washing in TBS-T for 3 × 10 min, the membranes were incubated with secondary antibodies for 1 h, washed again 3 × 10 min in TBS-T and developed using Clarity™ Western ECL Substrate (170-5061, BioRad). The chemiluminescent signal was detected using the ChemiDoc™ XRS+ imaging system from BioRad and the band intensities were quantified using ImageLab™ software (BioRad).

**Sample preparation for LC-MS.** Protein concentrations were quantified after lysis using the BCA according to manufacturer's instructions and the same amount of protein was used for sample preparation. Samples were reduced with 25% TFE and 20 mM TCEP at 55 °C for 20 min, followed by alkylation with 55 mM of 2-chloroacetamide (CAA) (C0267, Sigma) in the dark at room temperature for 30 min. Samples were digested with LysC (1:25 enzyme to protein ratio, Wako Chemicals Ltd), for 4–6 h before adding trypsin (1:25, Promega) for overnight digestion at 37 °C. The samples were dried by a centrifugal vacuum evaporator and desalted with Oasis HLB 96-well plate following the manufacturer's instructions. The desalted peptides were re-solubilized in 100 mM TEAB to 1 μg/μl. All the peptides were labeled with Isobaric Tandem Mass Tags -10plex TMT according to the manufacturer's protocol (90110, Thermo Scientific). The labeling was done at room temperature for at least 1 h and labeled samples were quenched using 10 μl of 1 M Tris (pH 7.4) solution. A high pH reverse phase Zorbax 300 Extend C-18 4.6 mm × 250 mm (Agilent) column and liquid chromatography AKTA Micro (GE) system was used for offline sample prefractionation. The fractions were concatenated into 20 fractions and dried with a centrifugal vacuum evaporator.

**LC-MS.** The digested, labeled, and dried peptide sample fractions were resuspended in 0.1% acetonitrile, 0.5% (v/v) acetic acid and 0.06% TFA in water immediately before analysis on LC-MS. Online chromatography was performed using Dionex UltiMate 3000 UPLC system coupled to a Q Exactive mass spectrometer (Thermo Scientific). Each fraction was separated on a 50 cm × 75 μm (ID) EASY-Spray analytical

column (ES903, Thermo Scientific) in a 80 min gradient of programmed mixture of solvent A (0.1% formic acid in $H_2O$) and solvent B (99.9% acetonitrile, 0.1% formic acid). MS data were acquired using a top 12 data-dependent acquisition method. Full scan MS spectra were acquired in the range of 350–1550 m/z at a resolution of 60,000 and AGC target of 3e6; Top 12 dd-MS2 60,000 and 1e5 with isolation window at 1.0 m/z.

## Quantification and statistical analysis

**Protein identification and quantification.** Protein identification was performed by Proteome Discoverer 2.5 software (Thermo Scientific), using both Mascot 2.6.0 (Matrix Science) and Sequest HT (Thermo Scientific) search engines to search against reviewed human Uniprot databases (downloaded on 13 Jan 2017, including 42,105 sequence entries and another downloaded on 23 Jul 2018, including 9606 sequence entries). MS precursor mass tolerance was set at 20ppm, fragment mass tolerance 0.05 Da, and maximum missed cleavage sites of 3. Dynamic modifications searched for Oxidation (M), Deamidation (NQ), and Acetylation (N-terminal protein). Static modifications: Carbamidomethyl (C) and TMT10plex (K and peptide N terminus). Only the spectrum peaks with signal-to-noise ratio (S/N) > 4 were chosen for searches. The false discovery rate (FDR) was set to 1% at both PSM and peptide levels. Only the unique and razor peptides were used for protein assignment and abundance quantification. Isotopic correction of the reporter ions in each TMT channel was performed according to the product sheet. Only the master proteins in the protein group were used for downstream analysis. For some datasets, the peptide abundances were obtained from Proteome Discoverer software (version 2.5). Every peptide with another modification than a TMT one was removed. To ensure the accuracy of TMT quantification, reporter S/N threshold was set at 10 and co-isolation threshold at 30%. Then, every peptide dataset has been treated the same way as the protein dataset according to the same method described in Dai et al.[23]. To illustrate the RESP effect, we summed the non Log2 transformed fold changes from the peptides located before the cleaved site and the ones after the cleaved site. Then those fold changes were Log2 transformed and plotted as bar plots.

**Quantitative MS data analysis and visualization.** Quantified protein/peptide abundances were imported into the R environment (http://www.R-project.org/) to facilitate the data analysis and visualization. Only the proteins with at least two quantifying abundance counts were used for downstream analysis. Data cleaning, normalization, and calculations of protein abundance and thermal stability differences in each condition were performed using the IMPRINTS.CETSA and the IMPRINTS.CETSA.app R packages[38].

**Protein-protein interaction network and gene ontology (GO) enrichment analysis.** Protein-protein interaction network for hits was obtained by importing the hitlist Uniprot IDs into Cytoscape v.3.9.1 (http://cytoscape.org). Using the embedded STRING interaction database (http://apps.cytoscape.org/apps/stringApp), a default confidence cut-off score of 0.4 was applied to retrieve the network. Each node represents one hit protein, and edges symbolize protein-protein interactions. Nodes explanation can be found on figure legends. Comparative GO analysis was performed using the ClueGO v2.5.1 plugin in Cytoscape (http://apps.cytoscape.org/apps/cluego). Hitlist Uniprot IDs were imported to query the GO-Biological Processes database (EBI-QuickGO-GOA-15783 terms/pathways with 17268 available unique genes-20.11.2017). The parameters for analysis were set as follows: Evidence code – All; Use Go Term Fusion; GO tree interval – Level 3–8; GO Term/Pathway Selection – Minimum 3 genes and threshold of 4% of genes per term; GO term connectivity threshold (Kappa score) – 0.4; Two-sided hypergeometric test with Bonferroni step down p-value correction. Only GO terms with p-value < 0.05 are shown. GO terms are presented as nodes and clustered together based on term similarity. Node size is proportional to the p-value for GO term enrichment. Node colors are set according to the treatment condition showing the % of visible proteins of a term/pathway.

**Data analysis and visualization.** All graphs were generated using GraphPad Prism, R environment, cytoscape or Biorender. All data are presented as mean with error bars representing the standard error of the mean (SEM). Error bars that are smaller than the displayed data points are not displayed by the software. Details regarding replicates for each experiment can be found in the figure legends. Sigmoidal curves were fit (where appropriate) using R environment. Unpaired t-tests were performed using GraphPad Prism and the results are displayed in figures and figure legends. The flow cytometry data was analyzed on FlowJo v10.8 and GraphPad Prism was used to represent the data.

## Reporting summary

Further information on research design is available in the Nature Portfolio Reporting Summary linked to this article.

## Data availability

All mass spectrometry raw data generated in this study have been deposited to the ProteomeXchange Consortium (http://proteomecentral.proteomexchange.org/) via the jPOST repository with the dataset identifiers PXD054912, PXD054911, PXD054910, PXD054909, PXD054908, PXD054907, PXD054903, PXD054902, PXD054901, PXD054854, PXD054853, PXD054852, PXD055016, PXD055015. Data cleaning, normalization, and calculations of protein abundance and thermal stability differences in each condition were performed using the IMPRINTS.CETSA and the IMPRINTS.CETSA.app R packages[38]. Source data are provided with this paper.

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

## Acknowledgements

We gratefully acknowledge funding from the Swedish Research Council (P.N.), the Swedish Cancer Society (P.N.), Radiumhemmet's funds (P.N.), the Knut and Alice Wallenberg Foundation (P.N.), Singapore's National Research Foundation (NRF-NRFI08-2022, NRF-CRP22-2019-0003 (W.L.T., N.P.)), Singapore's National Medical Research Council (MOH-001332-00 (Y.Y.L.)), Agency for Science, Technology and Research, Singapore, and the Singapore Ministry of Education under its Research Centres of Excellence initiative. We also acknowledge all past members of the P.N. lab.

## Author contributions

Conceptualization: W.L.T., P.N. and N.P.; Methodology: Y.Y.L., L.H.V., K.K., M.R., H.M.T., W.L.T., P.N. and N.P.; Formal Analysis: Y.Y.L., L.H.V., K.K., M.R., M.A.G., W.L.T., P.N. and N.P.; Investigation: Y.Y.L., L.H.V., K.K., H.M.T., M.R., J.J.H.L., J.L., A.C., A.D.J., W.L.T., P.N. and N.P.; Writing–Original draft: Y.Y.L., W.L.T., P.N. and N.P.; Writing–Review & Editing: All; Funding Acquisition: P.N., W.L.T. and N.P.; Supervision: A.D.J., P.N., W.L.T., N.P.

## Funding

## Competing interests
