## [Transparent Peer Review file · Nature Communications]

MS CETSA deep functional proteomics uncovers novel DNA repair programs leading to Gemcitabine resistance

Corresponding Author: Dr Pär Nordlund

Version 0:

Reviewer comments:

Reviewer #1

(Remarks to the Author)

Innate and acquired drug resistance is a major challenge in cancer therapy, driven by genetic or epigenetic changes that rewire key cellular pathways like apoptosis or DNA repair. While genomics and transcriptomics have previously been used to identify resistance mechanisms, these methods are often insufficient to provide a systems-level understanding. In this manuscript, Liang et al. take an innovative approach by using MS-CETSA to investigate drug resistance mechanisms in both cancer cell lines and patient samples. The authors previously developed MS-CETSA, which measures drug-protein and protein-protein interactions over time, serving as a proxy for pathway modulation and cell state changes.

Here, they focus on the cellular response to gemcitabine, a widely used cancer drug, in resistant and sensitive diffuse large B cell lymphoma cell lines. Using IMPRINTS-CETSA, they show that the initial response (3 hrs) to gemcitabine is similar in both resistant and sensitive cell lines, with RRM1 stabilisation, consistent with gemcitabine's role as an RNR inhibitor, and RPA complex stabilisation at replication forks, leading to ATR-CHK1 activation. At later time points (5-8 hrs), the responses diverge: sensitive cells activate apoptosis, while resistant cells activate cell cycle progression (CDKs) and TLS bypass of gemcitabine lesions. They also identify a novel DNA repair protein ensemble, termed Auxiliary DNA Damage Repair (ADDR), which promotes drug resistance to gemcitabine and other DNA-damaging agents.

Given the gemcitabine-induced ATR activation, the authors demonstrate that inhibiting ATR in resistant cells destabilises ADDR and TLS factors, resulting in drug hypersensitivity and overcoming resistance. This aligns with ongoing clinical trials of ATR inhibitors combined with gemcitabine, providing insight into the synthetic lethality observed in these treatments. Finally, the authors briefly show that their CETSA method can be used to analyse drug responses in patient samples, identifying apoptotic gene activation linked to drug sensitivity.

This is a well-written and original manuscript. The use of IMPRINTS-CETSA to explore drug resistance mechanisms in cancer is promising and could inform decisions on personalised treatments for resistant cancers. However, several points need to be addressed before publication.

Major Points

1. Although the manuscript aims to use MS-CETSA to identify therapy resistance in patients, it only analyses samples from two gemcitabine-naive patients. The authors should expand their study to include drug-resistant cancer samples to better demonstrate the ability to identify resistance mechanisms.
2. Given that resistance in cancer is often heterogeneous, the authors need to explain how MS-CETSA can be interpreted when analysing complex cell populations from patients with multiple resistance mechanisms.
3. MS-CETSA may overlook resistance mechanisms caused by mutations or downregulation in genes involved in nucleoside analogue activation (e.g., nucleoside transporters or kinases), which could prevent downstream drug engagement. A combination of MS-CETSA with genomics and transcriptomics would provide a more comprehensive view of resistance mechanisms. The authors should comment on this limitation.
4. Why is RRM1/2 engagement/stabilisation only observed in resistant cells at 8 hours (Fig 3B)?

Minor Points

1. The methodology is complex, particularly for non-experts. A figure illustrating the key concepts of CETSA and its integration with protein interaction networks would be helpful (e.g., modifying Fig 1A to better explain the CETSA concept such as protein stabilisation/destabilisation).
2. The manuscript uses too many acronyms and initialisms, making it harder to follow. The authors should consider simplifying them (e.g., IMPRINTS-CETSA, MS-CETSA, ITDR-CETSA, IMPRINTS, PRINTS, CCAE, ADDR, RESP). For instance, the authors have described "Protein Interaction States" in previous papers without using the acronym "PRINTS."

Reviewer #2

(Remarks to the Author)

In this work, Liang and colleagues use MS-CETSA to profile the response of four diffuse large B-cell lymphoma (DLBCL) cell lines to gemcitabine in a time-resolved manner. This included two cell lines that are drug-sensitive and two that are drug-resistant. At early time points, the authors identify DNA damage response proteins with altered thermal stability in all cell lines in agreement with the appearance of stalled replication forks upon treatment with gemcitabine. At later time points, gemcitabine was shown to induce apoptosis in sensitive cells, while resistant cells progressed in their cell cycle, likely through activation of translesion synthesis polymerases. Inhibition of the translesion synthesis pathway was able to partially re-sensitize cells to gemcitabine, as was inhibition of ATR (a kinase involved in sensing DNA damage). In parallel, the authors identify a group of proteins that they term auxiliary DNA damage repair proteins (ADDR) that might be involved in a more general resistance mechanism to DNA damaging drugs. Overall, this manuscript describes a thorough and impressive characterization of the mechanism of action and resistance of gemcitabine and provides hints to revert resistance. I have no major comments to the methodology used, and have only some suggestions that could further improve the manuscript:

1. It would be interesting to profile a bit further the cell lines used. Are there any mutations and/or baseline proteome changes between the cell lines that could hint at the effects seen? Particularly, why are these cell lines able to induce ADDR and TLS, while the sensitive cells are not?
2. Since the manuscript describing the CETSA apoptosis ensemble (CCAЕ) of proteins is not available, the authors could provide that list of proteins as a supplementary table or pre-print the other manuscript. Can anything be learned from the specific proteins in this CCAЕ that change (n=24) vs the ones that do not change (n=23) in sensitive cells? Is there a particular apoptosis-inducing drug in the other manuscript that gives this particular signal?

Best regards,
André Mateus

Reviewer #3

(Remarks to the Author)

In this study, Liang et al. employed Integrated Modulation of Protein Interaction States-Cellular Thermal Shift Assay (IMPRINT-CETSA), a system-wide functional proteomic approach, to investigate cellular resistance to gemcitabine in diffuse large B-cell lymphoma (DLBCL) cell lines. The authors found that gemcitabine activates similar proteomic responses across all tested DLBCL cell lines. Notably, by 5-8 hours, the responses in sensitive and resistant cell lines start to bifurcate, where sensitive cell lines activate apoptosis, indicated by 24 hits in the core CETSA apoptosis ensemble (CCAЕ). In contrast, resistant cell lines exhibit a translesion synthesis (TLS) response, represented by an increase in the thermal stability and abundance of TLS-related proteins. Furthermore, the authors identified an auxiliary DNA damage repair (ADDR) response protein ensemble that is specifically activated in resistant cells. The authors proposed that both TLS and ADDR responses promote ATR-dependent drug resistance. Finally, they showed the use of CETSA profiles to predict drug sensitivity in patients

In summary, this study uses time-dependent CETSA to investigate differential drug responses, providing a system-level profiling of gemcitabine-induced proteomic responses with mechanistical insights. However, earlier studies on gemcitabine resistance mechanisms (PMID: 24376779, 29849128, 16603639) and ATR-induced TLS response upon DNA damage (PMID: 31189884, 39215012) limit the novelty of the current work. Moreover, some of the conclusions and claims require additional evidence and explanations. Below are some suggestions to strengthen the manuscript.

Major points

1. The interpretations of thermal stability shifts require more experimental evidence and literature supports. For example, in Figure 2, replication protein A subunits (RPA1, RPA2, and RPA3) exhibit thermal stabilization upon gemcitabine treatment. How can this phenomenon imply ssDNA binding? Furthermore, in Figure 5, the thermal destabilization of Polδ is interpreted as an outcome of its release from replication fork. How can these changes in protein thermal stability be linked to specific biochemical events?
2. To categorize thermal stability shift, it would be helpful to have a quantitative metric or at least provide a solid workflow for how the classification is made. The illustration in Figure 1C seems to suggest that "temperature-dependency" is taking into account for defining CETSA destabilization or stabilization. If that is the case, why the authors interpret the CETSA profile of CHEK1 in Figure 2 as thermal destabilization while its CETSA profile seems to be complex and could be hard to define. For example, the thermal stability decreases first and then increases as temperature rises for 1 hour, 48 nM gemcitabine treatment. Similarly, how do the authors interpret CETSA profile of DNMT1 in sensitive cells in Figure 2?

3. In Figure 2, the CETSA profile of DNMT1 protein seems to differ between sensitive and resistant cells. What's the explanation for this difference?

4. The results in Figure 2 indicate that in addition to the known gemcitabine target, RRM1, CETSA can detect the activation of ATR/CHEK1 and alteration in DNMT1 as general effects of gemcitabine. It is not clear how the authors reason that the resistance mechanisms occur downstream of these events (lines 181-183).

Minor points:

1. Is there any data for RPA3 treated with 280 nM gemcitabine?

2. The plot is misaligned for PCLAF in Figure 5 (SUDHL4, 24 uM gemcitabine, 8h).

3. The interpretation of the CETSA profile in Figure 1C needs to be improved and more detailed.

4. If the authors only want to address the early response (1hr) in Fig. 1D, it may not be necessary to include results from other timepoints.

5. Figure 2 presentation needs to be improved. The current order of figure layout is different from the order of description in the main text.

6. The results in Figure 5C are unclear. What are the molecular weights of Polk and its degraded form? The staining is quite weak. How do the authors determine which bands correspond to the cleaved Polk?

7. Throughout the whole article, there isn't any explanation of the asterisk labels regarding the statistical significance. Also, in Figure 5B, what are the groups being compared in the statistical tests are not mentioned.

Version 1:

Reviewer comments:

Reviewer #1

(Remarks to the Author)

I am satisfied with the manuscript improvements and replies to my comments and happy to support publication.

Reviewer #2

(Remarks to the Author)

The authors have addressed my suggestions. I congratulate the authors on the nice work!

Reviewer #3

(Remarks to the Author)

The authors have addressed most of my questions.

Reviewer #1 (Remarks to the Author):

Innate and acquired drug resistance is a major challenge in cancer therapy, driven by genetic or epigenetic changes that rewire key cellular pathways like apoptosis or DNA repair. While genomics and transcriptomics have previously been used to identify resistance mechanisms, these methods are often insufficient to provide a systems-level understanding. In this manuscript, Liang et al. take an innovative approach by using MS-CETSA to investigate drug resistance mechanisms in both cancer cell lines and patient samples. The authors previously developed MS-CETSA, which measures drug-protein and protein-protein interactions over time, serving as a proxy for pathway modulation and cell state changes.

Here, they focus on the cellular response to gemcitabine, a widely used cancer drug, in resistant and sensitive diffuse large B cell lymphoma cell lines. Using IMPRINTS-CETSA, they show that the initial response (3 hrs) to gemcitabine is similar in both resistant and sensitive cell lines, with RRM1 stabilisation, consistent with gemcitabine's role as an RNR inhibitor, and RPA complex stabilisation at replication forks, leading to ATR-CHK1 activation. At later time points (5-8 hrs), the responses diverge: sensitive cells activate apoptosis, while resistant cells activate cell cycle progression (CDKs) and TLS bypass of gemcitabine lesions. They also identify a novel DNA repair protein ensemble, termed Auxiliary DNA Damage Repair (ADDR), which promotes drug resistance to gemcitabine and other DNA-damaging agents.

Given the gemcitabine-induced ATR activation, the authors demonstrate that inhibiting ATR in resistant cells destabilises ADDR and TLS factors, resulting in drug hypersensitivity and overcoming resistance. This aligns with ongoing clinical trials of ATR inhibitors combined with gemcitabine, providing insight into the synthetic lethality observed in these treatments. Finally, the authors briefly show that their CETSA method can be used to analyse drug responses in patient samples, identifying apoptotic gene activation linked to drug sensitivity.

This is a well-written and original manuscript. The use of IMPRINTS-CETSA to explore drug resistance mechanisms in cancer is promising and could inform decisions on personalised treatments for resistant cancers. However, several points need to be addressed before publication.

Major Points

1. Although the manuscript aims to use MS-CETSA to identify therapy resistance in patients, it only analyses samples from two gemcitabine-naive patients. The authors should expand their study to include drug-resistant cancer samples to better demonstrate the ability to identify resistance mechanisms.

We agree with the reviewer that adding more clinical data could be very interesting. However, due to the focus of the work on defining the mechanism for gemcitabine resistance in DLBCL, and the large bulk of data already in the manuscript, we feel a more extensive characterization of effects in clinical samples is outside the scope of this first study. Nevertheless, we think that the identified resistance mechanism is very likely relevant for the clinical situation, as the used cell lines are directly patient derived cells (in contrast to many other mechanistic studies with generated cell lines, where the resistance has been evolved in the lab). Also, the fact that we identify the same resistance mechanism in cell lines derived from two different patients, support that this is indeed a program that is in effect in a clinical situation. As a note, if we wanted to include additional gemcitabine DLBCL patient samples that are gemcitabine treated and resistant, 1) these would be rare cases since Gemcitabine is not the first line of treatment and 2) a biopsy is rarely collected in such a case.

2. Given that resistance in cancer is often heterogeneous, the authors need to explain how MS-CETSA can be interpreted when analysing complex cell populations from patients with multiple resistance mechanisms.

The unique strength of the MS-CETSA technology lies in the fact that changes in protein interaction states are measured on a proteome-wide scale, therefore multiple resistance mechanisms can potentially be assessed in the same experiment. In this study we focused on an overlapping program in the two resistant cell lines (which are derived from two different patients), although there might be additional induced and well-defined programs present in each cell line. Therefore, we envisage that studies of further patient derived cell lines by us and others will allow for more induced resistance programs to be identified (e.g. by overlapping responses) to establish a repertoire of available resistance programs for gemcitabine in DLBCL. It remains to be seen if this repertoire of biochemical resistance programs is small (and can be saturated) or large, but our impression from this study and other unpublished studies in our lab is that there are some dominant programs that are seen in multiple cell lines. Based on the repertoire of programs, we envisage that in real heterogeneous patient samples, although each program is just present in low to medium stoichiometry (20-50%), it will be possible to identify which programs are dominating in the sample, to potentially guide therapeutic decisions. After such a treatment, new resistance programs might be enriched and should be accessible to detect with CETSA. We have now added a comment on this issue in the manuscript (refer to pg no. in MS).

3. MS-CETSA may overlook resistance mechanisms caused by mutations or downregulation in genes involved in nucleoside analogue activation (e.g., nucleoside transporters or kinases), which could prevent downstream drug

engagement. A combination of MS-CETSA with genomics and transcriptomics would provide a more comprehensive view of resistance mechanisms. The authors should comment on this limitation.

We have now added an analysis of mutations as well as quantitative proteomics of the cell line (SFig 8C-F) and conclude that the information content on resistance mechanisms in such static omics data is relatively sparse as compared to time dependent CETSA data.

Considering drug target engagement, the comparable isothermal dose response curves in sensitive and resistant cells treated with gemcitabine (Fig 1E) suggest similar drug target engagement between these cells. Additionally, this study includes one dataset that accurately recapitulates the reviewer's described scenario: a key player of the nucleotide metabolism, SAMHD1, is knocked out, thus possibly preventing downstream drug target engagement. In these cells the gemcitabine-induced thermal stabilization of RRM1 was abolished (SFig 5F), highlighting that MS-CETSA can detect such loss of drug target engagement.

4. Why is RRM1/2 engagement/stabilisation only observed in resistant cells at 8 hours (Fig 3B)?

We have now clarified that at 8h only the catalytic subunit of RNR, RRM1 is shifting in both sensitive and resistant cells (grey node) (SFig 8B), which was already observed as early as 1h and in all cell lines (Fig 1D). The radical subunit, RRM2, appeared as hit in response to 8h gemcitabine treatment in resistant cells (SFig 8B, green node) through its level changes, which we described later as part of ADDR ensemble (Fig 5A).

Minor Points

1. The methodology is complex, particularly for non-experts. A figure illustrating the key concepts of CETSA and its integration with protein interaction networks would be helpful (e.g., modifying Fig 1A to better explain the CETSA concept such as protein stabilisation/destabilisation).

We have now modified Figure 1 to include an illustration of the concept of CETSA thermal stabilization/destabilization and how this translates to the IMPRINTS bar graphs shown and discussed in most of the following figures.

2. The manuscript uses too many acronyms and initialisms, making it harder to follow. The authors should consider simplifying them (e.g., IMPRINTS-CETSA, MS-CETSA, ITDR-CETSA, IMPRINTS, PRINTS, CCAE, ADDR, RESP). For instance, the authors have described "Protein Interaction States" in previous papers without using the acronym "PRINTS."

We have now simplified the text.

Reviewer #2 (Remarks to the Author):

In this work, Liang and colleagues use MS-CETSA to profile the response of four diffuse large B-cell lymphoma (DLBCL) cell lines to gemcitabine in a time-resolved manner. This included two cell lines that are drug-sensitive and two that are drug-resistant. At early time points, the authors identify DNA damage response proteins with altered thermal stability in all cell lines in agreement with the appearance of stalled replication forks upon treatment with gemcitabine. At later time points, gemcitabine was shown to induce apoptosis in sensitive cells, while resistant cells progressed in their cell cycle, likely through activation of translesion synthesis polymerases. Inhibition of the translesion synthesis pathway was able to partially re-sensitize cells to gemcitabine, as was inhibition of ATR (a kinase involved in sensing DNA damage). In parallel, the authors identify a group of proteins that they term auxiliary DNA damage repair proteins (ADDR) that might be involved in a more general resistance mechanism to DNA damaging drugs. Overall, this manuscript describes a thorough and impressive characterization of the mechanism of action and resistance of gemcitabine and provides hints to revert resistance. I have no major comments to the methodology used, and have only some suggestions that could further improve the manuscript:

1. It would be interesting to profile a bit further the cell lines used. Are there any mutations and/or baseline proteome changes between the cell lines that could hint at the effects seen? Particularly, why are these cell lines able to induce ADDR and TLS, while the sensitive cells are not?

As mentioned in response to reviewer 1 we have now added and analyzed mutational data of these cell lines as well as added static quantitative proteomics data of the 4 cell lines (baseline proteomics data)(SFig 8C-F). However, the GO analysis of these data do not give any clues to a resistance mechanism.

2. Since the manuscript describing the CETSA apoptosis ensemble (CCAIE) of proteins is not available, the authors could provide that list of proteins as a supplementary table or pre-print the other manuscript. Can anything be learned from the specific proteins in this CCAIE that change (n=24) vs the ones that do not change (n=23) in sensitive cells? Is there a particular apoptosis-inducing drug in the other manuscript that gives this particular signal?

The manuscript defining the CCAIE motif is now out, Ramos et al Cell Reports, 2024;43(10):114784. Some of the proteins that are not part of the overlap is due to them not being well measured or that shifts are slightly below the cutoff. To allow the reader to access this for each protein we have now included the CCAIE IMPRINTS profiles comparing the responses at 8h gemcitabine treatment in sensitive and resistant versus the Ramos et al CCAIE proteins (SFig 3). There might be smaller effects in the gemcitabine data for chromatin proteins and mitochondrial proteins, but due to the low numbers it is hard to associate this to a distinct mechanistic difference.

Best regards,
André Mateus

Reviewer #3 (Remarks to the Author):

In this study, Liang et al. employed Integrated Modulation of Protein Interaction States-Cellular Thermal Shift Assay (IMPRINT-CETSA), a system-wide functional proteomic approach, to investigate cellular resistance to gemcitabine in diffuse large B-cell lymphoma (DLBCL) cell lines. The authors found that gemcitabine activates similar proteomic responses across all tested DLBCL cell lines. Notably, by 5-8 hours, the responses in sensitive and resistant cell lines start to bifurcate, where sensitive cell lines activate apoptosis, indicated by 24 hits in the core CETSA apoptosis ensemble (CCAЕ). In contrast, resistant cell lines exhibit a translesion synthesis (TLS) response, represented by an increase in the thermal stability and abundance of TLS-related proteins. Furthermore, the authors identified an auxiliary DNA damage repair (ADDR) response protein ensemble that is specifically activated in resistant cells. The authors proposed that both TLS and ADDR responses promote ATR-dependent drug resistance. Finally, they showed the use of CETSA profiles to predict drug sensitivity in patients

In summary, this study uses time-dependent CETSA to investigate differential drug responses, providing a system-level profiling of gemcitabine-induced proteomic responses with mechanistical insights. However, earlier studies on gemcitabine resistance mechanisms (PMID: 24376779, 29849128, 16603639) and ATR-induced TLS response upon DNA damage (PMID: 31189884, 39215012) limit the novelty of the current work. Moreover, some of the conclusions and claims require additional evidence and explanations. Below are some suggestions to strengthen the manuscript.

Major points

1. The interpretations of thermal stability shifts require more experimental evidence and literature supports. For example, in Figure 2, replication protein A subunits (RPA1, RPA2, and RPA3) exhibit thermal stabilization upon gemcitabine treatment. How can this phenomenon imply ssDNA binding? Furthermore, in Figure 5, the thermal destabilization of Pol δ is interpreted as an outcome of its release from replication fork. How can these changes in protein thermal stability be linked to specific biochemical events?

CETSA shifts strongly support changes of protein interaction states through interactions with e.g. a protein, a nucleic acid or a metabolite, or a PTM change, in the cell at a certain time point. However, the exact molecular origin of the stability effect is challenging to access experimentally and will in many cases remain tentative and be defined by the known interaction made by a protein. As an example, even though we have shown that the destabilisation of CHEK1 is concomitant with phosphorylation, i.e. activation, we don't know which specific interactions are changed to induce this destabilisation. Could be primarily interactions around the phosphorylated residue but could also be interactions with other proteins, etc. So in essence we relate the shifts to a functional state of a protein (activated Chek1, single strand bound RPA, activated CDK1, Dai Cell 2018 173:1481, etc), rather than prove the exact molecular background for the change. We feel, for understanding cellular process, the possibility to follow transitions between different functional states of proteins at the proteome-level is of great value, as shown in the present study. There are a few instances where statements around states might not have been clear so we have now reformulated a few sentences in the text in line with this.

2. To categorize thermal stability shift, it would be helpful to have a quantitative metric or at least provide a solid workflow for how the classification is made. The illustration in Figure 1C seems to suggest that "temperature-dependency" is taking into account for defining CETSA destabilization or stabilization. If that is the case, why the authors interpret the CETSA profile of CHEK1 in Figure 2 as thermal destabilization while its CETSA profile seems to be complex and could be hard to define. For example, the thermal stability decreases first and then increases as temperature rises for 1 hour, 48 nM gemcitabine treatment. Similarly, how do the authors interpret CETSA profile of DNMT1 in sensitive cells in Figure 2?

As discussed in response to reviewer 1, we have now modified Figure 1 to include an illustration of the concept of CETSA thermal stabilisation/destabilisation and how this translates to the IMPRINTS bar graphs shown and discussed in most of the following figures.

The proteins in Figure 2 are based on being common hits in resistant and sensitive cells with similar CETSA profiles across the various time points, albeit with different intensities where some (e.g. CHEK1 at 1h) don't pass hit selection criteria).

The CETSA profile for DNMT1 in sensitive cells in Figure 2 with only effect at the higher temperature is a bit different to the resistant cells, but its presence in resistant cells give support for, at least at the late timepoint, there is a very significant shift in DNMT1 in sensitive cells at the highest temperature. DNMT1 shifts are further discussed in the next question below.

3. In Figure 2, the CETSA profile of DNMT1 protein seems to differ between sensitive and resistant cells. What's the explanation for this difference?

We agree with the reviewer that there appears to be a very significant difference in the profiles in sensitive and resistant cells for DNMT1. This could be interesting and inform on structural/interaction differences of the protein

in the two types of cells. However, when, as mentioned above, the molecular causes for the shifts are hard to follow up, we selected to not address this difference specifically in the text. Conceptually, there is a possibility that the different profiles reflect a difference in the protein population/states that shifts, or potentially (but less likely, when the profile follows the phenotype), that the intrinsic stability of the DNMT1 protein is different in the different strains, effecting the temperature range in which the protein shifts due to interaction changes.

4. The results in Figure 2 indicate that in addition to the known gemcitabine target, RRM1, CETSA can detect the activation of ATR/CHEK1 and alteration in DNMT1 as general effects of gemcitabine. It is not clear how the authors reason that the resistance mechanisms occur downstream of these events (lines 181-183).

We have now added a brief schematic to clarify this (SFig 9) where, in resistant cells the activation ATR subsequently activate a signaling pathway (as stated in Discussion section, details remain to be elucidated) that lead to the hallmarks of the resistance response, the ADDR, TLS and the opening of cell cycle checkpoints. This signaling pathway does not seem to be operative in sensitive cells, or alternatively, apoptosis is happening in parallel and attenuates this pathway.

Minor points:

1. Is there any data for RPA3 treated with 280 nM gemcitabine?

The RPA3 data for 280nM gemcitabine condition was incomplete with missing quantitative measurements for some conditions and was therefore removed during our data analysis cleaning step.

2. The plot is misaligned for PCLAF in Figure5 (SUDHL4, 24 uM gemcitabine, 8h).

The Figure has been adjusted

3. The interpretation of the CETSA profile in Figure 1C needs to be improved and more detailed.

We have now modified Figure 1 to include an illustration of the concept of CETSA thermal stabilisation/destabilisation and how this translates to the IMPRINTS bar graphs shown and discussed in most of the following figures.

4. If the authors only want to address the early response (1hr) in Fig. 1D, it may not be necessary to include results from other timepoints.

The additional timepoints are intentionally depicted to demonstrate the consistent CETSA profiles over time.

5. Figure 2 presentation needs to be improved. The current order of figure layout is different from the order of description in the main text.

We have matched the figure layout with the description in the main text.

6. The results in Figure 5C are unclear. What are the molecular weights of Polk and its degraded form? The staining is quite weak. How the authors determine which bands correspond to the cleaved Polk?

We have now replaced image with a clearer one and moved this section to supplementary data (SFig 6). The observed molecular weights of the degraded form of PolK were ~65-70kDa. Since their appearance and disappearance correlated with gemcitabine and zVAD-FMK treatment, respectively, we judged these bands as specific for PolK. However, as we did not further confirm and validate these bands as cleaved PolK, we have now removed this statement from the manuscript and instead only focus on the main PolK band where we now included a quantification showing the level decrease and rescue by zVAD-FMK.

7. Throughout the whole article, there isn't any explanation of the asterisk labels regarding the statistical significance. Also, in Figure 5B, what are the groups being compared in the statistic tests are not mentioned.

Figure legends have been improved to include type of statistical tests and compared groups.